

# Adaptations of data mining methodologies: a systematic literature review

Veronika Plotnikova, Marlon Dumas and Fredrik Milani

Institute of Computer Science, University of Tartu, Tartu, Estonia

## ABSTRACT

The use of end-to-end data mining methodologies such as CRISP-DM, KDD process, and SEMMA has grown substantially over the past decade. However, little is known as to how these methodologies are used in practice. In particular, the question of whether data mining methodologies are used 'as-is' or adapted for specific purposes, has not been thoroughly investigated. This article addresses this gap via a systematic literature review focused on the context in which data mining methodologies are used and the adaptations they undergo. The literature review covers 207 peer-reviewed and 'grey' publications. We find that data mining methodologies are primarily applied 'as-is'. At the same time, we also identify various adaptations of data mining methodologies and we note that their number is growing rapidly. The dominant adaptations pattern is related to methodology adjustments at a granular level (modifications) followed by extensions of existing methodologies with additional elements. Further, we identify two recurrent purposes for adaptation: (1) adaptations to handle Big Data technologies, tools and environments (technological adaptations); and (2) adaptations for context-awareness and for integrating data mining solutions into business processes and IT systems (organizational adaptations). The study suggests that standard data mining methodologies do not pay sufficient attention to deployment issues, which play a prominent role when turning data mining models into software products that are integrated into the IT architectures and business processes of organizations. We conclude that refinements of existing methodologies aimed at combining data, technological, and organizational aspects, could help to mitigate these gaps.

## INTRODUCTION

The availability of Big Data has stimulated widespread adoption of data mining and data analytics in research and in business settings (*Columbus, 2017*). Over the years, a certain number of data mining methodologies have been proposed, and these are being used extensively in practice and in research. However, little is known about what and how data mining methodologies are applied, and it has not been neither widely researched nor discussed. Further, there is no consolidated view on what constitutes quality of methodological process in data mining and data analytics, how data mining and data analytics are applied/used in organization settings context, and how application practices relate to each other. That motivates the need for comprehensive survey in the field.

Corresponding author
Veronika Plotnikova,
veronika.plotnikova@ut.ee

There have been surveys or quasi-surveys and summaries conducted in related fields. Notably, there have been two systematic systematic literature reviews; Systematic Literature Review, hereinafter, SLR is the most suitable and widely used research method for identifying, evaluating and interpreting research of particular research question, topic or phenomenon (*Kitchenham, Budgen & Brereton, 2015*). These reviews concerned Big Data Analytics, but not general purpose data mining methodologies. *Adrian et al. (2004)* executed SLR with respect to implementation of Big Data Analytics (BDA), specifically, capability components necessary for BDA value discovery and realization. The authors identified BDA implementation studies, determined their main focus areas, and discussed in detail BDA applications and capability components. *Saltz & Shamshurin (2016)* have published SLR paper on Big Data Team Process Methodologies. Authors have identified lack of standard in regards to how Big Data projects are executed, highlighted growing research in this area and potential benefits of such process standard. Additionally, authors synthesized and produced list of 33 most important success factors for executing Big Data activities. Finally, there are studies that surveyed data mining techniques and applications across domains, yet, they focus on data mining process artifacts and outcomes (*Madni, Anwar & Shah, 2017*; *Liao, Chu & Hsiao, 2012*), but not on end-to-end process methodology.

There have been number of surveys conducted in domain-specific settings such as hospitality, accounting, education, manufacturing, and banking fields. *Mariani et al. (2018)* focused on Business Intelligence (BI) and Big Data SLR in the hospitality and tourism environment context. *Amani & Fadlalla (2017)* explored application of data mining methods in accounting while *Romero & Ventura (2013)* investigated educational data mining. Similarly, *Hassani, Huang & Silva (2018)* addressed data mining application case studies in banking and explored them by three dimensions—topics, applied techniques and software. All studies were performed by the means of systematic literature reviews. Lastly, *Bi & Cochran (2014)* have undertaken standard literature review of Big Data Analytics and its applications in manufacturing.

Apart from domain-specific studies, there have been very few general purpose surveys with comprehensive overview of existing data mining methodologies, classifying and contextualizing them. Valuable synthesis was presented by *Kurgan & Musilek (2006)* as comparative study of the state-of-the art of data mining methodologies. The study was not SLR, and focused on comprehensive comparison of phases, processes, activities of data mining methodologies; application aspect was summarized briefly as application statistics by industries and citations. Three more comparative, non-SLR studies were undertaken by *Marban, Mariscal & Segovia (2009)*, *Mariscal, Marbán & Fernández (2010)*, and the most recent and closest one by *Martnez-Plumed et al. (2017)*. They followed the same pattern with systematization of existing data mining frameworks based on comparative analysis. There, the purpose and context of consolidation was even more practical—to support derivation and proposal of the new artifact, that is, novel data mining methodology. The majority of the given general type surveys in the field are more than a decade old, and have natural limitations due to being: (1) non-SLR studies, and

(2) so far restricted to comparing methodologies in terms of phases, activities, and other elements.

The key common characteristic behind all the given studies is that data mining methodologies are treated as normative and standardized ('one-size-fits-all') processes. A complementary perspective, not considered in the above studies, is that data mining methodologies are not normative standardized processes, but instead, they are frameworks that need to be specialized to different industry domains, organizational contexts, and business objectives. In the last few years, a number of extensions and adaptations of data mining methodologies have emerged, which suggest that existing methodologies are not sufficient to cover the needs of all application domains. In particular, extensions of data mining methodologies have been proposed in the medical domain (*Niaksu, 2015*), educational domain (*Tavares, Vieira & Pedro, 2017*), the industrial engineering domain (*Huber et al., 2019*; *Solarte, 2002*), and software engineering (*Marbán et al., 2007*, *2009*). However, little attention has been given to studying how data mining methodologies are applied and used in industry settings, so far only non-scientific practitioners' surveys provide such evidence.

Given this research gap, the central objective of this article is to investigate how data mining methodologies are applied by researchers and practitioners, both in their generic (standardized) form and in specialized settings. This is achieved by investigating if data mining methodologies are applied 'as-is' or adapted, and for what purposes such adaptations are implemented.

Guided by Systematic Literature Review method, initially we identified a corpus of primary studies covering both peer-reviewed and 'grey' literature from 1997 to 2018. An analysis of these studies led us to a taxonomy of uses of data mining methodologies, focusing on the distinction between 'as is' usage versus various types of methodology adaptations. By analyzing different types of methodology adaptations, this article identifies potential gaps in standard data mining methodologies both at the technological and at the organizational levels.

The rest of the article is organized as follows. The *Background* section provides an overview of key concepts of data mining and associated methodologies. Next, *Research Design* describes the research methodology. The *Findings and Discussion* section presents the study results and their associated interpretation. Finally, threats to validity are addressed in *Threats to Validity* while the *Conclusion* summarizes the findings and outlines directions for future work.

## BACKGROUND

The section introduces main data mining concepts, provides overview of existing data mining methodologies, and their evolution.

Data mining is defined as a set of rules, processes, algorithms that are designed to generate actionable insights, extract patterns, and identify relationships from large datasets (*Morabito, 2016*). Data mining incorporates automated data extraction, processing, and modeling by means of a range of methods and techniques. In contrast, data

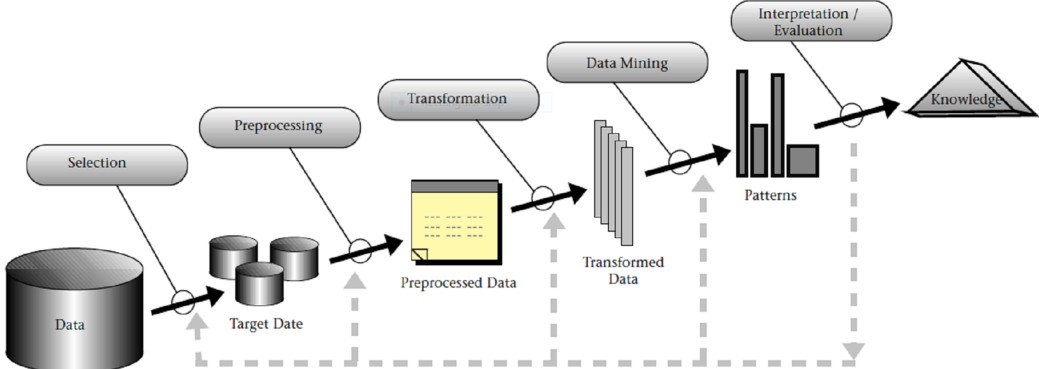

**Figure 1 An overview of the steps composing the KDD process, as presented in** *Fayyad, Piatetsky-Shapiro & Smyth (1996a, 1996c)*.

analytics refers to techniques used to analyze and acquire intelligence from data (including 'big data') (*Gandomi & Haider, 2015*) and is positioned as a broader field, encompassing a wider spectrum of methods that includes both statistical and data mining (*Chen, Chiang & Storey, 2012*). A number of algorithms has been developed in statistics, machine learning, and artificial intelligence domains to support and enable data mining. While statistical approaches precedes them, they inherently come with limitations, the most known being rigid data distribution conditions. Machine learning techniques gained popularity as they impose less restrictions while deriving understandable patterns from data (*Bose & Mahapatra, 2001*).

Data mining projects commonly follow a structured process or methodology as exemplified by *Mariscal, Marbán & Fernández (2010)*, *Marban, Mariscal & Segovia (2009)*. A data mining methodology specifies tasks, inputs, outputs, and provides guidelines and instructions on how the tasks are to be executed (*Mariscal, Marbán & Fernández, 2010*). Thus, data mining methodology provides a set of guidelines for executing a set of tasks to achieve the objectives of a data mining project (*Mariscal, Marbán & Fernández, 2010*).

The foundations of structured data mining methodologies were first proposed by *Fayyad, Piatetsky-Shapiro & Smyth (1996a, 1996b, 1996c)*, and were initially related to Knowledge Discovery in Databases (KDD). KDD presents a conceptual process model of computational theories and tools that support information extraction (knowledge) with data (*Fayyad, Piatetsky-Shapiro & Smyth, 1996a*). In KDD, the overall approach to knowledge discovery includes data mining as a specific step. As such, KDD, with its nine main steps (exhibited in Fig. 1), has the advantage of considering data storage and access, algorithm scaling, interpretation and visualization of results, and human computer interaction (*Fayyad, Piatetsky-Shapiro & Smyth, 1996a, 1996c*). Introduction of KDD also formalized clearer distinction between data mining and data analytics, as for example formulated in *Tsai et al. (2015)*: "…by the data analytics, we mean the whole KDD process, while by the data analysis, we mean the part of data analytics that is aimed at finding the hidden information in the data, such as data mining".

The main steps of KDD are as follows:

- Step 1: Learning application domain: In the first step, it is needed to develop an understanding of the application domain and relevant prior knowledge followed by identifying the goal of the KDD process from the customer's viewpoint.
- Step 2: Dataset creation: Second step involves selecting a dataset, focusing on a subset of variables or data samples on which discovery is to be performed.
- Step 3: Data cleaning and processing: In the third step, basic operations to remove noise or outliers are performed. Collection of necessary information to model or account for noise, deciding on strategies for handling missing data fields, and accounting for data types, schema, and mapping of missing and unknown values are also considered.
- Step 4: Data reduction and projection: Here, the work of finding useful features to represent the data, depending on the goal of the task, application of transformation methods to find optimal features set for the data is conducted.
- Step 5: Choosing the function of data mining: In the fifth step, the target outcome (e.g., summarization, classification, regression, clustering) are defined.
- Step 6: Choosing data mining algorithm: Sixth step concerns selecting method(s) to search for patterns in the data, deciding which models and parameters are appropriate and matching a particular data mining method with the overall criteria of the KDD process.
- Step 7: Data mining: In the seventh step, the work of mining the data that is, searching for patterns of interest in a particular representational form or a set of such representations: classification rules or trees, regression, clustering is conducted.
- Step 8: Interpretation: In this step, the redundant and irrelevant patterns are filtered out, relevant patterns are interpreted and visualized in such way as to make the result understandable to the users.
- Step 9: Using discovered knowledge: In the last step, the results are incorporated with the performance system, documented and reported to stakeholders, and used as basis for decisions.

The KDD process became dominant in industrial and academic domains (*Kurgan & Musilek, 2006*; *Marban, Mariscal & Segovia, 2009*). Also, as timeline-based evolution of data mining methodologies and process models shows (Fig. 2 below), the original KDD data mining model served as basis for other methodologies and process models, which addressed various gaps and deficiencies of original KDD process. These approaches extended the initial KDD framework, yet, extension degree has varied ranging from process restructuring to complete change in focus. For example, *Brachman & Anand (1996)* and further *Gertosio & Dussauchoy (2004)* (in a form of case study) introduced practical adjustments to the process based on iterative nature of process as well as interactivity. The complete KDD process in their view was enhanced with supplementary tasks and the focus was changed to user's point of view (human-centered approach), highlighting decisions that need to be made by the user in the course of data mining process. In contrast, *Cabena et al. (1997)* proposed different number of steps emphasizing

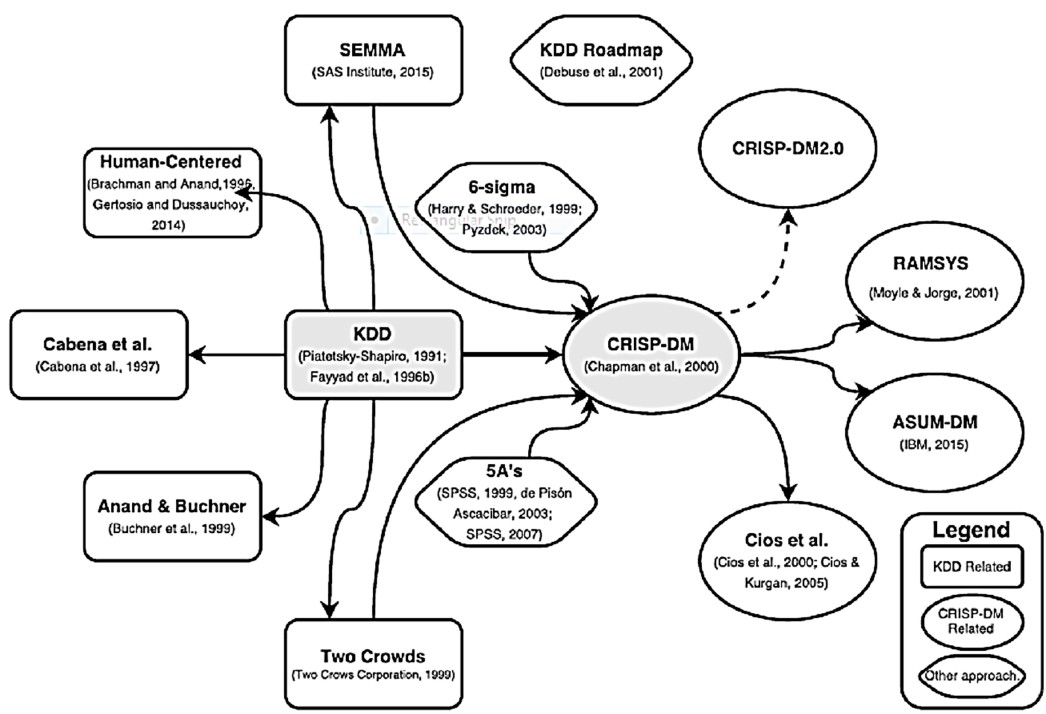

**Figure 2** Evolution of data mining process and methodologies, as presented in *Martnez-Plumed et al. (2017)*.

and detailing data processing and discovery tasks. Similarly, in a series of works *Anand & Büchner (1998)*, *Anand et al. (1998)*, *Buchner et al. (1999)* presented additional data mining process steps by concentrating on adaptation of data mining process to practical settings. They focused on cross-sales (entire life-cycles of online customer), with further incorporation of internet data discovery process (web-based mining). Further, Two Crows data mining process model is consultancy originated framework that has defined the steps differently, but is still close to original KDD. Finally, SEMMA (Sample, Explore, Modify, Model and Assess) based on KDD, was developed by SAS institute in 2005 (*SAS Institute Inc., 2017*). It is defined as a logical organization of the functional toolset of SAS Enterprise Miner for carrying out the core tasks of data mining. Compared to KDD, this is vendor-specific process model which limits its application in different environments. Also, it skips two steps of original KDD process ('Learning Application Domain' and 'Using of Discovered Knowledge') which are regarded as essential for success of data mining project (*Mariscal, Marbán & Fernández, 2010*). In terms of adoption, new KDD-based proposals received limited attention across academia and industry (*Kurgan & Musilek, 2006*; *Marban, Mariscal & Segovia, 2009*). Subsequently, most of these methodologies converged into the CRISP-DM methodology.

Additionally, there have only been two non-KDD based approaches proposed alongside extensions to KDD. The first one is 5A's approach presented by *De Pisón Ascacbar (2003)* and used by SPSS vendor. The key contribution of this approach has been related to adding 'Automate' step while disadvantage was associated with omitting 'Data Understanding' step. The second approach was 6-Sigma which is industry originated

method to improve quality and customer's satisfaction (*Pyzdek & Keller, 2003*). It has been successfully applied to data mining projects in conjunction with DMAIC performance improvement model (Define, Measure, Analyze, Improve, Control).

In 2000, as response to common issues and needs (*Marban, Mariscal & Segovia, 2009*), an industry-driven methodology called Cross-Industry Standard Process for Data Mining (CRISP-DM) was introduced as an alternative to KDD. It also consolidated original KDD model and its various extensions. While CRISP-DM builds upon KDD, it consists of six phases that are executed in iterations (*Marban, Mariscal & Segovia, 2009*). The iterative executions of CRISP-DM stand as the most distinguishing feature compared to initial KDD that assumes a sequential execution of its steps. CRISP-DM, much like KDD, aims at providing practitioners with guidelines to perform data mining on large datasets. However, CRISP-DM with its six main steps with a total of 24 tasks and outputs, is more refined as compared to KDD. The main steps of CRIPS-DM, as depicted in Fig. 3 below are as follows:

- Phase 1: Business understanding: The focus of the first step is to gain an understanding of the project objectives and requirements from a business perspective followed by converting these into data mining problem definitions. Presentation of a preliminary plan to achieve the objectives are also included in this first step.
- Phase 2: Data understanding: This step begins with an initial data collection and proceeds with activities in order to get familiar with the data, identify data quality issues, discover first insights into the data, and potentially detect and form hypotheses.
- Phase 3: Data preparation: The third step covers activities required to construct the final dataset from the initial raw data. Data preparation tasks are performed repeatedly.
- Phase 4: Modeling phase: In this step, various modeling techniques are selected and applied followed by calibrating their parameters. Typically, several techniques are used for the same data mining problem.
- Phase 5: Evaluation of the model(s): The fifth step begins with the quality perspective and then, before proceeding to final model deployment, ascertains that the model(s) achieves the business objectives. At the end of this phase, a decision should be reached on how to use data mining results.
- Phase 6: Deployment phase: In the final step, the models are deployed to enable end-customers to use the data as basis for decisions, or support in the business process. Even if the purpose of the model is to increase knowledge of the data, the knowledge gained will need to be organized, presented, distributed in a way that the end-user can use it. Depending on the requirements, the deployment phase can be as simple as generating a report or as complex as implementing a repeatable data mining process.

The development of CRISP-DM was led by industry consortium. It is designed to be domain-agnostic (*Mariscal, Marbán & Fernández, 2010*) and as such, is now widely used by industry and research communities (*Marban, Mariscal & Segovia, 2009*). These distinctive characteristics have made CRISP-DM to be considered as 'de-facto' standard of data mining methodology and as a reference framework to which other methodologies are benchmarked (*Mariscal, Marbán & Fernández, 2010*).
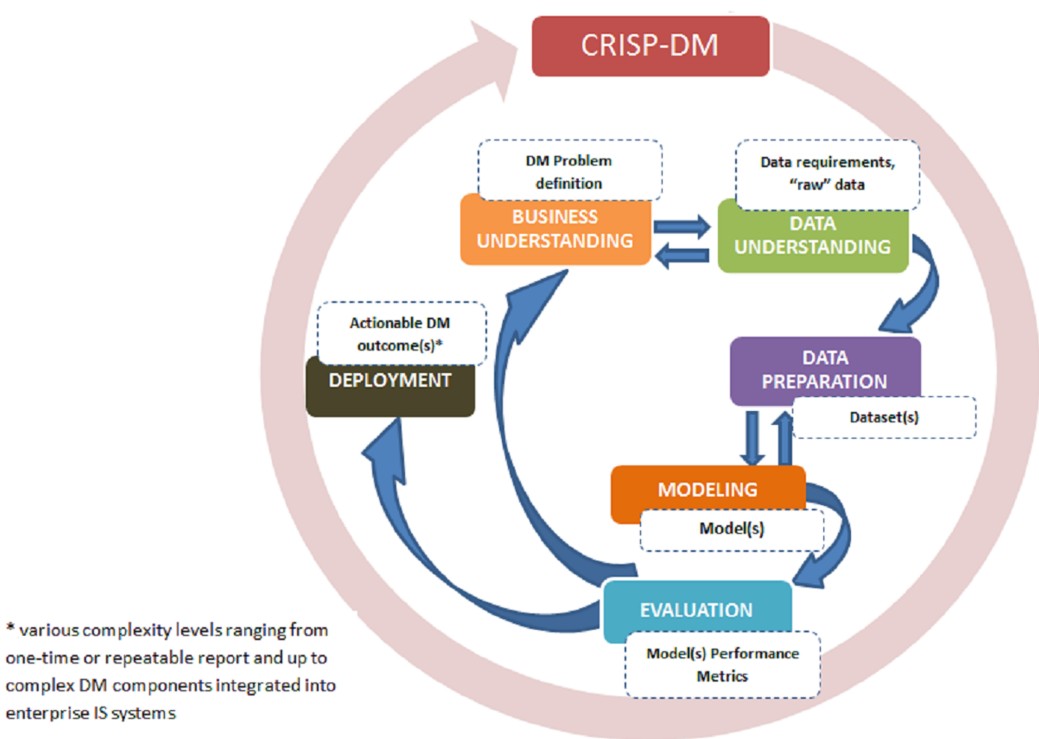

**Figure 3 CRISP-DM phases and key outputs (adapted from *Chapman et al. (2000)*).**

Similarly to KDD, a number of refinements and extensions of the CRISP-DM methodology have been proposed with the two main directions—extensions of the process model itself and adaptations, merger with the process models and methodologies in other domains. Extensions direction of process models could be exemplified by *Cios & Kurgan (2005)* who have proposed integrated Data Mining & Knowledge Discovery (DMKD) process model. It contains several explicit feedback mechanisms, modification of the last step to incorporate discovered knowledge and insights application as well as relies on technologies for results deployment. In the same vein, *Moyle & Jorge (2001)*, *Blockeel & Moyle (2002)* proposed Rapid Collaborative Data Mining System (RAMSYS) framework—this is both data mining methodology and system for remote collaborative data mining projects. The RAMSYS attempted to achieve the combination of a problem solving methodology, knowledge sharing, and ease of communication. It intended to allow the collaborative work of remotely placed data miners in a disciplined manner as regards information flow while allowing the free flow of ideas for problem solving (*Moyle & Jorge, 2001*). CRISP-DM modifications and integrations with other specific domains were proposed in Industrial Engineering (Data Mining for Industrial Engineering by *Solarte (2002)*), and Software Engineering by *Marbán et al. (2007, 2009)*. Both approaches enhanced CRISP-DM and contributed with additional phases, activities and tasks typical for engineering processes, addressing on-going support (*Solarte, 2002*), as well as project management, organizational and quality assurance tasks (*Marbán et al., 2009*).

**Table 1 Key aspects of existing data mining process models and methodologies.**

| Name | Origin | Basis | Key concept | Year |
|---|---|---|---|---|
| Human-Centered | Academy | KDD | Iterative process and interactivity (user's point of view and needed decisions) | 1996, 2004 |
| Cabena et al. | Academy | KDD | Focus on data processing and discovery tasks | 1997 |
| Anand and Buchner | Academy | KDD | Supplementary steps and integration of web-mining | 1998, 1999 |
| Two Crows | Industry | KDD | Modified definitions of steps | 1998 |
| SEMMA | Industry | KDD | Tool-specific (SAS Institute), elimination of some steps | 2005 |
| 5 A's | Industry | Independent | Supplementary steps | 2003 |
| 6 Sigmas | Industry | Independent | Six Sigma quality improvement paradigm in conjunction with DMAIC performance improvement model | 2003 |
| CRISP-DM | Joint industry and academy | KDD | Iterative execution of steps, significant refinements to tasks and outputs | 2000 |
| Cios et al. | Academy | Crisp-DM | Integration of data mining and knowledge discovery, feedback mechanisms, usage of received insights supported by technologies | 2005 |
| RAMSYS | Academy | Crisp-DM | Integration of collaborative work aspects | 2001–2002 |
| DMIE | Academy | Crisp-DM | Integration and adaptation to Industrial Engineering domain | 2001 |
| Marban | Academy | Crisp-DM | Integration and adaptation to Software Engineering domain | 2007 |
| KDD roadmap | Joint industry and academy | Independent | Tool-specific, resourcing task | 2001 |
| ASUM | Industry | Crisp-DM | Tool-specific, combination of traditional Crisp-DM and agile implementation approach | 2015 |

Finally, limited number of attempts to create independent or semi-dependent data mining frameworks was undertaken after CRISP-DM creation. These efforts were driven by industry players and comprised KDD Roadmap by *Debuse et al. (2001)* for proprietary predictive toolkit (Lanner Group), and recent effort by IBM with Analytics Solutions Unified Method for Data Mining (ASUM-DM) in 2015 (*IBM Corporation, 2016*: https://developer.ibm.com/technologies/artificial-intelligence/articles/architectural-thinking-in-the-wild-west-of-data-science/). Both frameworks contributed with additional tasks, for example, resourcing in KDD Roadmap, or hybrid approach assumed in ASUM, for example, combination of agile and traditional implementation principles.

The Table 1 above summarizes reviewed data mining process models and methodologies by their origin, basis and key concepts.

## RESEARCH DESIGN

The main research objective of this article is to study how data mining methodologies are applied by researchers and practitioners. To this end, we use systematic literature review (SLR) as scientific method for two reasons. Firstly, systematic review is based on trustworthy, rigorous, and auditable methodology. Secondly, SLR supports structured synthesis of existing evidence, identification of research gaps, and provides framework to position new research activities (*Kitchenham, Budgen & Brereton, 2015*). For our SLR, we followed the guidelines proposed by *Kitchenham, Budgen & Brereton (2015)*. All SLR details have been documented in the separate, peer-reviewed SLR protocol (available at https://figshare.com/articles/Systematic-Literature-Review-Protocol/10315961).

## Research questions

As suggested by *Kitchenham, Budgen & Brereton (2015)*, we have formulated research questions and motivate them as follows. In the preliminary phase of research we have discovered very limited number of studies investigating data mining methodologies application practices as such. Further, we have discovered number of surveys conducted in domain-specific settings, and very few general purpose surveys, but none of them considered application practices either. As contrasting trend, recent emergence of limited number of adaptation studies have clearly pinpointed the research gap existing in the area of application practices. Given this research gap, in-depth investigation of this phenomenon led us to ask: "How data mining methodologies are applied ('as-is' vs adapted) (RQ1)?" Further, as we intended to investigate in depth universe of adaptations scenarios, this naturally led us to RQ2: "How have existing data mining methodologies been adapted?" Finally, if adaptions are made, we wish to explore what the associated reasons and purposes are, which in turn led us to RQ3: "For what purposes are data mining methodologies adapted?"

Thus, for this review, there are three research questions defined:

- **Research Question 1: How data mining methodologies are applied ('as-is' versus adapted)?** This question aims to identify data mining methodologies application and usage patterns and trends.
- **Research Question 2: How have existing data mining methodologies been adapted?** This questions aims to identify and classify data mining methodologies adaptation patterns and scenarios.
- **Research Question 3: For what purposes have existing data mining methodologies been adapted?** This question aims to identify, explain, classify and produce insights on what are the reasons and what benefits are achieved by adaptations of existing data mining methodologies. Specifically, what gaps do these adaptations seek to fill and what have been the benefits of these adaptations. Such systematic evidence and insights will be valuable input to potentially new, refined data mining methodology. Insights will be of interest to practitioners and researchers.

## Data collection strategy

Our data collection and search strategy followed the guidelines proposed by *Kitchenham, Budgen & Brereton (2015)*. It defined the scope of the search, selection of literature and electronic databases, search terms and strings as well as screening procedures.

### Primary search

The primary search aimed to identify an initial set of papers. To this end, the search strings were derived from the research objective and research questions. The term 'data mining' was the key term, but we also included 'data analytics' to be consistent with observed research practices. The terms 'methodology' and 'framework' were also included.
Thus, the following search strings were developed and validated in accordance with the guidelines suggested by *Kitchenham, Budgen & Brereton (2015)*:

('data mining methodology') OR ('data mining framework') OR ('data analytics methodology') OR ('data analytics framework')

The search strings were applied to the indexed scientific databases Scopus, Web of Science (for 'peer-reviewed', academic literature) and to the non-indexed Google Scholar (for non-peer-reviewed, so-called 'grey' literature). The decision to cover 'grey' literature in this research was motivated as follows. As proposed in number of information systems and software engineering domain publications (*Garousi, Felderer & Mäntylä, 2019*; *Neto et al., 2019*), SLR as stand-alone method may not provide sufficient insight into 'state of practice'. It was also identified (*Garousi, Felderer & Mäntylä, 2016*) that 'grey' literature can give substantial benefits in certain areas of software engineering, in particular, when the topic of research is related to industrial and practical settings. Taking into consideration the research objectives, which is investigating data mining methodologies application practices, we have opted for inclusion of elements of Multivocal Literature Review (MLR)[1] in our study. Also, *Kitchenham, Budgen & Brereton (2015)* recommends including 'grey' literature to minimize publication bias as positive results and research outcomes are more likely to be published than negative ones. Following MLR practices, we also designed inclusion criteria for types of 'grey' literature reported below.

The selection of databases is motivated as follows. In case of peer-reviewed literature sources we concentrated to avoid potential omission bias. The latter is discussed in IS research (*Levy & Ellis, 2006*) in case research is concentrated in limited disciplinary data sources. Thus, broad selection of data sources including multidisciplinary-oriented (Scopus, Web of Science, Wiley Online Library) and domain-oriented (ACM Digital Library, IEEE Xplorer Digital Library) scientific electronic databases was evaluated. Multidisciplinary databases have been selected due to wider domain coverage and it was validated and confirmed that they do include publications originating from domain-oriented databases, such as ACM and IEEE. From multi-disciplinary databases as such, Scopus was selected due to widest possible coverage (it is worlds largest database, covering app. 80% of all international peer-reviewed journals) while Web of Science was selected due to its longer temporal range. Thus, both databases complement each other. The selected non-indexed database source for 'grey' literature is Google Scholar, as it is comprehensive source of both academic and 'grey' literature publications and referred as such extensively (*Garousi, Felderer & Mäntylä, 2019*; *Neto et al., 2019*).

Further, *Garousi, Felderer & Mäntylä (2019)* presented three-tier categorization framework for types of 'grey literature'. In our study we restricted ourselves to the 1st tier 'grey' literature publications of the limited number of 'grey' literature producers. In particular, from the list of producers (*Neto et al., 2019*) we have adopted and focused on government departments and agencies, non-profit economic, trade organizations ('think-tanks') and professional associations, academic and research institutions, businesses and corporations (consultancy companies and established private companies). The 1st tier 'grey' literature selected items include: (1) government, academic, and private sector consultancy

[1] Multivocal Literature Review (MLR) (as in *Garousi, Felderer & Mäntylä (2019)*), is a form of a SLR which includes the gray literature (e.g., blog posts, videos and white papers) in addition to the published (formal) literature (e.g., journal and conference papers).

2 Including white papers, market reports, industry overviews and similar.

reports[2], (2) theses (not lower than Master level) and PhD Dissertations, (3) research reports, (4) working papers, (5) conference proceedings, preprints. With inclusion of the 1st tier 'grey' literature criteria we mitigate quality assessment challenge especially relevant and reported for it (*Garousi, Felderer & Mäntylä, 2019*; *Neto et al., 2019*).

### Scope and domains inclusion

As recommended by *Kitchenham, Budgen & Brereton (2015)* it is necessary to initially define research scope. To clarify the scope, we defined what is not included and is out of scope of this research. The following aspects are not included in the scope of our study:

1. Context of technology and infrastructure for data mining/data analytics tasks and projects.
2. Granular methods application in data mining process itself or their application for data mining tasks, for example, constructing business queries or applying regression or neural networks modeling techniques to solve classification problems. Studies with granular methods are included in primary texts corpus as long as method application is part of overall methodological approach.
3. Technological aspects in data mining for example, data engineering, dataflows and workflows.
4. Traditional statistical methods not associated with data mining directly including statistical control methods.

Similarly to *Budgen et al. (2006)* and *Levy & Ellis (2006)*, initial piloting revealed that search engines retrieved literature available for all major scientific domains including ones outside authors' area of expertise (e.g., medicine). Even though such studies could be retrieved, it would be impossible for us to analyze and correctly interpret literature published outside the possessed area of expertise. The adjustments toward search strategy were undertaken by retaining domains closely associated with Information Systems, Software Engineering research. Thus, for Scopus database the final set of inclusive domains was limited to nine and included Computer Science, Engineering, Mathematics, Business, Management and Accounting, Decision Science, Economics, Econometrics and Finance, and Multidisciplinary as well as Undefined studies. Excluded domains covered 11.5% or 106 out of 925 publications; it was confirmed in validation process that they primarily focused on specific case studies in fundamental sciences and medicine[3]. The included domains from Scopus database were mapped to Web of Science to ensure consistent approach across databases and the correctness of mapping was validated.

3 Excluded domains were Medicine, Biochemistry, Genetics and Molecular Biology, Environmental Science, Earth and Planetary Science, Physics and Astronomy, Energy and Material Science, Agricultural and Biological Science, Chemistry and Chemical Engineering, Pharmacology, Toxicology and Pharmaceuticals, Arts and Humanities, Neuroscience, Immunology and Microbiology, Health Professions and Nursing.

### Screening criteria and procedures

Based on the SLR practices (as in *Kitchenham, Budgen & Brereton (2015)*, *Brereton et al. (2007)*) and defined SLR scope, we designed multi-step screening procedures (quality and relevancy) with associated set of *Screening Criteria* and *Scoring System*. The purpose of relevancy screening is to find relevant primary studies in an unbiased way (*Vanwersch et al., 2011*). Quality screening, on the other hand, aims to assess primary relevant studies in terms of quality in unbiased way.

*Screening Criteria* consisted of two subsets—*Exclusion Criteria* applied for initial filtering and *Relevance Criteria*, also known as *Inclusion Criteria*.

*Exclusion Criteria* were initial threshold quality controls aiming at eliminating studies with limited or no scientific contribution. The exclusion criteria also address issues of understandability, accessability and availability. The *Exclusion Criteria* were as follows:

1. Quality 1: The publication item is not in English (understandability).
2. Quality 2: Publication item duplicates which can occur when:

   - either the same document retrieved from two or all three databases.
   - or different versions of the same publication are retrieved (i.e., the same study published in different sources)—based on best practices, decision rule is that the most recent paper is retained as well as the one with the highest score (*Kofod-Petersen, 2014*).
   - if a publication is published both as conference proceeding and as journal article with the same name and same authors or as an extended version of conference paper, the latter is selected.

3. Quality 3: Length of the publication is less than 6 pages—short papers do not have the space to expand and discuss presented ideas in sufficient depth to examine for us.
4. Quality 4: The paper is not accessible in full length online through the university subscription of databases and via Google Scholar—not full availability prevents us from assessing and analyzing the text.

The initially retrieved list of papers was filtered based on *Exclusion Criteria*. Only papers that passed all criteria were retained in the final studies corpus. Mapping of criteria towards screening steps is exhibited in Fig. 4.

*Relevance Criteria* were designed to identify relevant publications and are presented in Table 2 below while mapping to respective process steps is presented in Fig. 4. These criteria were applied iteratively.

As a final SLR step, the full texts quality assessment was performed with constructed *Scoring Metrics* (in line with *Kitchenham & Charters (2007)*). It is presented in the Table 3 below.

## Data extraction and screening process

The conducted data extraction and screening process is presented in Fig. 4. In Step 1 initial publications list were retrieved from pre-defined databases—Scopus, Web of Science, Google Scholar. The lists were merged and duplicates eliminated in Step 2. Afterwards, texts being less than 6 pages were excluded (Step 3). Steps 1–3 were guided by *Exclusion Criteria*. In the next stage (Step 4), publications were screened by Title based on pre-defined *Relevance Criteria*. The ones which passed were evaluated by their availability (Step 5). As long as study was available, it was evaluated again by the same pre-defined *Relevance Criteria* applied to Abstract, Conclusion and if necessary Introduction (Step 6). The ones which passed this threshold formed primary publications corpus extracted from

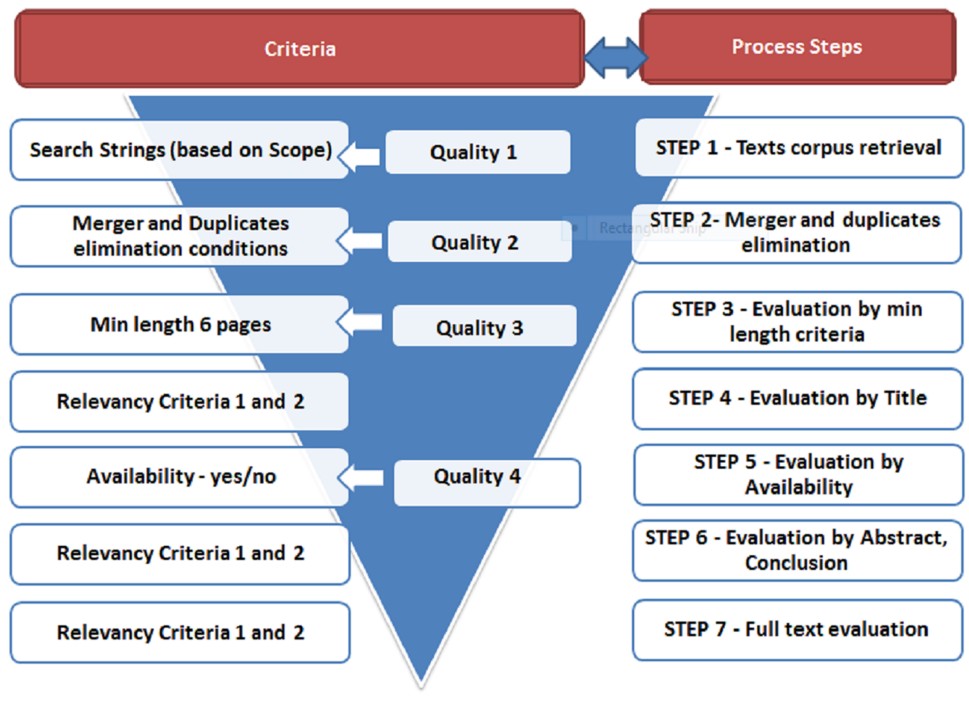

**Figure 4 Relevance and quality screening steps with criteria.**

**Table 2 Relevance criteria mapping to screening process steps.**

| Relevance criteria | Criteria definition | Criteria justification |
|---|---|---|
| Relevance 1 | Is the study about data mining or data analytics approach and is within designated list of domains? | Exclude studies conducted outside the designated domain list. Exclude studies not directly describing and/or discussing data mining and data analytics |
| Relevance 2 | Is the study introducing/describing data mining or data analytics methodology/framework or modifying existing approaches? | Exclude texts considering only specific, granular data mining and data analytics techniques, methods or traditional statistical methods. Exclude publications focusing on specific, granular data mining and data analytics process/sub-process aspects. Exclude texts where description and discussion of data mining methodologies or frameworks is manifestly missing |

databases in full. These primary texts were evaluated again based on full text (Step 7) applying *Relevance Criteria* first and then *Scoring Metrics*.

## Results and quantitative analysis

In Step 1, 1,715 publications were extracted from relevant databases with the following composition—Scopus (819), Web of Science (489), Google Scholar (407). In terms of scientific publication domains, Computer Science (42.4%), Engineering (20.6%), Mathematics (11.1%) accounted for app. 74% of Scopus originated texts. The same applies to Web of Science harvest. *Exclusion Criteria* application produced the following results.

**Table 3 Scoring metrics.**

| Score | Criteria definition |
|---|---|
| 3 | Data mining methodology or framework is presented in full. All steps described and explained, tests performed, results compared and evaluated. There is clear proposal on usage, application, deployment of solution in organization's business process(es) and IT/IS system, and/or prototype or full solution implementation is discussed. Success factors described and presented |
| 2 | Data mining methodology or framework is presented, some process steps are missing, but they do not impact the holistic view and understanding of the performed work. Data mining process is clearly presented and described, tests performed, results compared and evaluated. There is proposal on usage, application, deployment of solution in organization's business process(es) and IT/IS system(s) |
| 1 | Data mining methodology or framework is not presented in full, some key phases and process steps are missing. Publication focuses on one or some aspects (e.g., method, technique) |
| 0 | Data mining methodology or framework not presented as holistic approach, but on fragmented basis, study limited to some aspects (e.g., method or technique discussion, etc.) |

In Step 2, after eliminating duplicates, 1,186 texts were passed for minimum length evaluation, and 767 reached assessment by *Relevancy Criteria*.

As mentioned *Relevance Criteria* were applied iteratively (Step 4–6) and in conjunction with availability assessment. As a result, only 298 texts were retained for full evaluation with 241 originating from scientific databases while 57 were 'grey'. These studies formed primary texts corpus which was extracted, read in full and evaluated by *Relevance Criteria* combined with *Scoring Metrics*. The decision rule was set as follows. Studies that scored "1" or "0" were rejected, while texts with "3" and "2" evaluation were admitted as final primary studies corpus. To this end, as an outcome of SLR-based, broad, cross-domain publications collection and screening we identified 207 relevant publications from peer-reviewed (156 texts) and 'grey' literature (51 texts). Figure 5 below exhibits yearly published research numbers with the breakdown by 'peer-reviewed' and 'grey' literature starting from 1997.

In terms of composition, 'peer-reviewed' studies corpus is well-balanced with 72 journal articles and 82 conference papers while book chapters account for 4 instances only. In contrast, in 'grey' literature subset, articles in moderated and non-peer reviewed journals are dominant ($n = 34$) compared to overall number of conference papers ($n = 13$), followed by small number of technical reports and pre-prints ($n = 4$).

Temporal analysis of texts corpus (as per Fig. 5 below) resulted in two observations. Firstly, we note that stable and significant research interest (in terms of numbers) on data mining methodologies application has started around a decade ago—in 2007. Research efforts made prior to 2007 were relatively limited with number of publications below 10. Secondly, we note that research on data mining methodologies has grown substantially since 2007, an observation supported by the 3-year and 10-year constructed mean trendlines. In particular, the number of publications have roughly tripled over past decade hitting all time high with 24 texts released in 2017.

Further, there are also two distinct spike sub-periods in the years 2007–2009 and 2014–2017 followed by stable pattern with overall higher number of released publications on annual basis. This observation is in line with the trend of increased penetration of methodologies, tools, cross-industry applications and academic research of data mining.

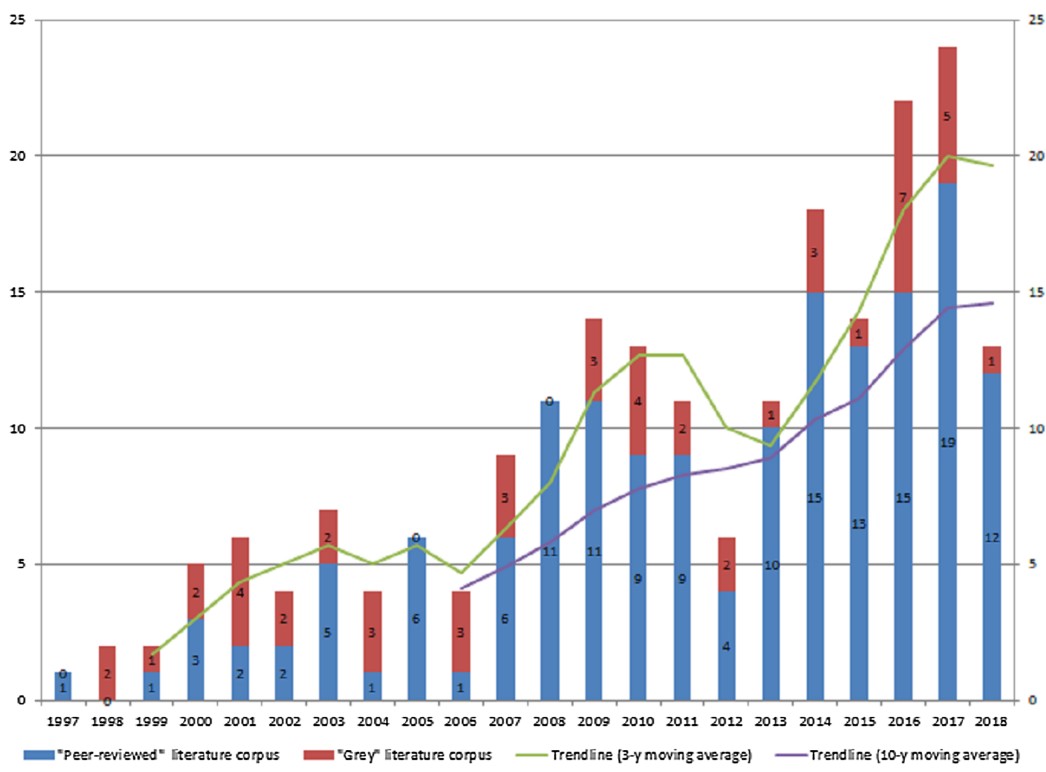

**Figure 5** SLR derived relevant texts corpus—data mining methodologies peer-reviewed research and 'grey' for period 1997–2018 (no. of publications).

## FINDINGS AND DISCUSSION

In this section, we address the research questions of the paper. Initially, as part of RQ1, we present overview of data mining methodologies 'as-is' and adaptation trends. In addressing RQ2, we further classify the adaptations identified. Then, as part of RQ3 subsection, each category identified under RQ2 is analyzed with particular focus on the goals of adaptations.

### RQ1: How data mining methodologies are applied ('as-is' vs. adapted)?

The first research question examines the extent to which data mining methodologies are used 'as-is' versus adapted. Our review based on 207 publications identified two distinct paradigms on how data mining methodologies are applied. The first is 'as-is' where the data mining methodologies are applied as stipulated. The second is with 'adaptations'; that is, methodologies are modified by introducing various changes to the standard process model when applied.

We have aggregated research by decades to differentiate application pattern between two time periods 1997–2007 with limited vs 2008–2018 with more intensive data mining application. The given cut has not only been guided by extracted publications corpus but also by earlier surveys. In particular, during the pre-2007 research, there where ten new methodologies proposed, but since then, only two new methodologies have been

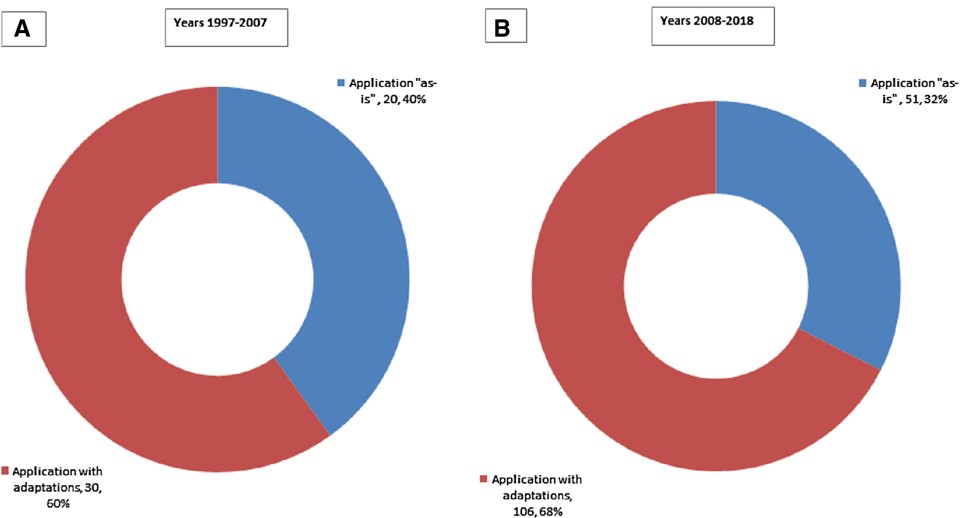

**Figure 6 Applications of data mining methodologies: (A) breakdown by 'as-is' vs. adaptions for 1997–2007 period; (B) breakdown by 'as-is' vs. adaptions for 2008–2018 period.**

proposed. Thus, there is a distinct trend observed over the last decade of large number of extensions and adaptations proposed vs entirely new methodologies.

We note that during the first decade of our time scope (1997–2007), the ratio of data mining methodologies applied 'as-is' was 40% (as presented in Fig. 6A). However, the same ratio for the following decade is 32% (Fig. 6B). Thus, in terms of relative shares we note a clear decrease in using data mining methodologies 'as-is' in favor of adapting them to cater to specific needs.The trend is even more pronounced when comparing numbers— adaptations more than tripled (from 30 to 106) while 'as-is' scenario has increased modestly (from 20 to 51). Given this finding, we continue with analyzing how data mining methodologies have been adapted under RQ2.

## RQ2: How have existing data mining methodologies been adapted?

We identified that data mining methodologies have been adapted to cater to specific needs. In order to categorize adaptations scenarios, we applied a two-level dichotomy, specifically, by applying the following decision tree:

1. Level 1 Decision: Has the methodology been combined with another methodology? If yes, the resulting methodology was classified in the 'integration' category. Otherwise, we posed the next question.
2. Level 2 Decision: Are any new elements (phases, tasks, deliverables) added to the methodology? If yes, we designate the resulting methodology as an 'extension' of the original one. Otherwise, we classify the resulting methodology as a modification of the original one.

Thus, when adapted three distinct types of adaptation scenarios can be distinguished:

- Scenario 'Modification': introduces specialized sub-tasks and deliverables in order to address specific use cases or business problems. Modifications typically concentrate

on granular adjustments to the methodology at the level of sub-phases, tasks or deliverables within the existing reference frameworks (e.g., CRISP-DM or KDD) stages. For example, *Chernov et al. (2014)*, in the study of mobile network domain, proposed automated decision-making enhancement in the deployment phase. In addition, the evaluation phase was modified by using both conventional and own-developed performance metrics. Further, in a study performed within the financial services domain, *Yang et al. (2016)* presents feature transformation and feature selection as sub-phases, thereby enhancing the data mining modeling stage.

- Scenario 'Extension': primarily proposes significant extensions to reference data mining methodologies. Such extensions result in either integrated data mining solutions, data mining frameworks serving as a component or tool for automated IS systems, or their transformations to fit specialized environments. The main purposes of extensions are to integrate fully-scaled data mining solutions into IS/IT systems and business processes and provide broader context with useful architectures, algorithms, etc. Adaptations, where extensions have been made, elicit and explicitly present various artifacts in the form of system and model architectures, process views, workflows, and implementation aspects. A number of soft goals are also achieved, providing holistic perspective on data mining process, and contextualizing with organizational needs. Also, there are extensions in this scenario where data mining process methodologies are substantially changed and extended in all key phases to enable execution of data mining life-cycle with the new (Big) Data technologies, tools and in new prototyping and deployment environments (e.g., Hadoop platforms or real-time customer interfaces). For example, *Kisilevich, Keim & Rokach (2013)* presented extensions to traditional CRISP-DM data mining outcomes with fully fledged Decision Support System (DSS) for hotel brokerage business. Authors (*Kisilevich, Keim & Rokach, 2013*) have introduced spatial/non-spatial data management (extending data preparation), analytical and spatial modeling capabilities (extending modeling phase), provided spatial display and reporting capabilities (enhancing deployment phase). In the same work domain knowledge was introduced in all phases of data mining process, and usability and ease of use were also addressed.

- Scenario 'Integration': combines reference methodology, for example, CRISP-DM with: (1) data mining methodologies originated from other domains (e.g., Software engineering development methodologies), (2) organizational frameworks (Balanced Scorecard, Analytics Canvass, etc.), or (3) adjustments to accommodate Big Data technologies and tools. Also, adaptations in the form of 'Integration' typically introduce various types of ontologies and ontology-based tools, domain knowledge, software engineering, and BI-driven framework elements. Fundamental data mining process adjustments to new types of data, IS architectures (e.g., real time data, multi-layer IS) are also presented. Key gaps addressed with such adjustments are prescriptive nature and low degree of formalization in CRISP-DM, obsolete nature of CRISP-DM with respect to tools, and lack of CRISP-DM integration with other organizational frameworks. For example, *Brisson & Collard (2008)* developed KEOPS data mining methodology

(CRIPS-DM based) centered on domain knowledge integration. Ontology-driven information system has been proposed with integration and enhancements to all steps of data mining process. Further, an integrated expert knowledge used in all data mining phases was proved to produce value in data mining process.

To examine how the application scenario of each data mining methodology usage has developed over time, we mapped peer-reviewed texts and 'grey' literature to respective adaptation scenarios, aggregated by decades (as presented in the Fig. 7 for peer-reviewed and Fig. 8 for 'grey').

For peer-reviewed research, such temporal analysis resulted in three observations. Firstly, research efforts in each adaptation scenario has been growing and number of publication more than quadrupled (128 vs. 28). Secondly, as noted above relative proportion of 'as-is' studies is diluted (from 39% to 33%) and primarily replaced with 'Extension' paradigm (from 25% to 30%). In contrast, in relative terms 'Modification' and 'Integration' paradigms gains are modest. Further, this finding is reinforced with other observation—most notable gaps in terms of modest number of publications remain in 'Integration' category where excluding 2008–2009 spike, research efforts are limited and number of texts is just 13. This is in stark contrast with prolific research in 'Extension category' though concentrated in the recent years. We can hypothesize that existing reference methodologies do not accommodate and support increasing complexity of data mining projects and IS/IT infrastructure, as well as certain domains specifics and as such need to be adapted.

In 'grey' literature, in contrast to peer-reviewed research, growth in number of publications is less profound—29 vs. 22 publications or 32% comparing across two decade (as per Fig. 8). The growth is solely driven by 'Integration' scenarios application (13 vs. 4 publications) while both 'as-is' and other adaptations scenarios are stagnating or in decline.

### RQ3: For what purposes have existing data mining methodologies been adapted?

We address the third research question by analyzing what gaps the data mining methodology adaptations seek to fill and the benefits of such adaptations. We identified three adaptation scenarios, namely 'Modification', 'Extension', and 'Integration'. Here, we analyze each of them.

### Modification

Modifications of data mining methodologies are present in 30 peer-reviewed and 4 'grey' literature studies. The analysis shows that modifications overwhelmingly consist of specific case studies. However, the major differentiating point compared to 'as-is' case studies is clear presence of specific adjustments towards standard data mining process methodologies. Yet, the proposed modifications and their purposes do not go beyond traditional data mining methodologies phases. They are granular, specialized and executed on tasks, sub-tasks, and at deliverables level. With modifications, authors describe

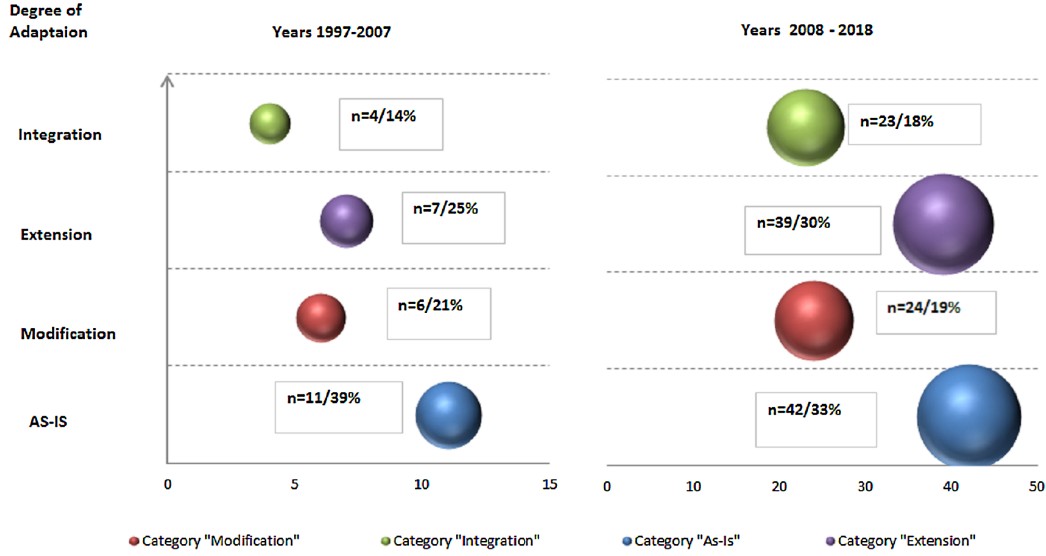

**Figure 7 Data Mining methodologies application research—primary 'peer-reviewed' texts classification by types of scenarios aggregated by decades (with numbers and relative proportions).**

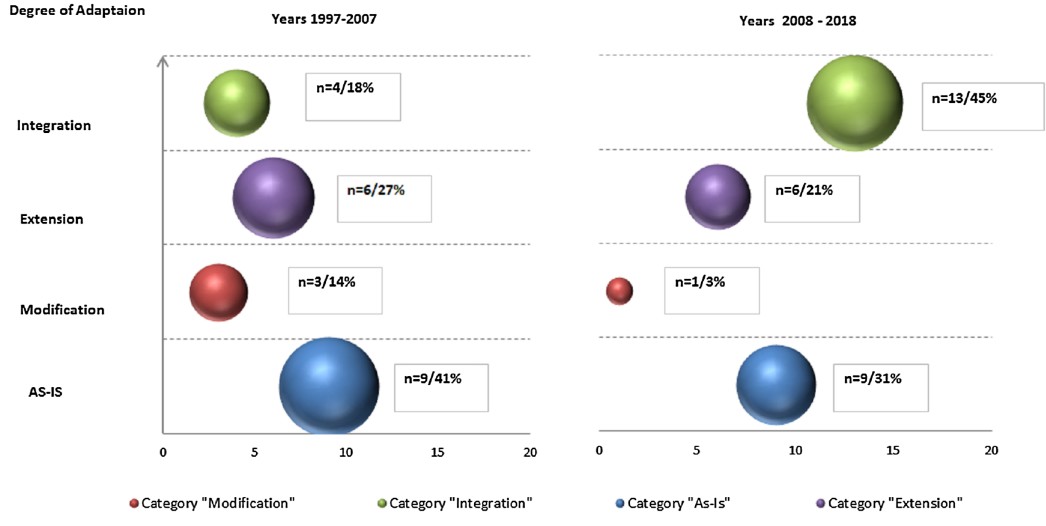

**Figure 8 Data Mining methodologies application research—primary 'grey' texts classification by types of scenarios aggregated by decades (with numbers and relative proportions).**

potential business applications and deployment scenarios at a conceptual level, but typically do not report or present real implementations in the IS/IT systems and business processes.

Further, this research subcategory can be best classified based on domains where case studies were performed and data mining methodologies modification scenarios executed. We have identified four distinct domain-driven applications presented in the Fig. 9.

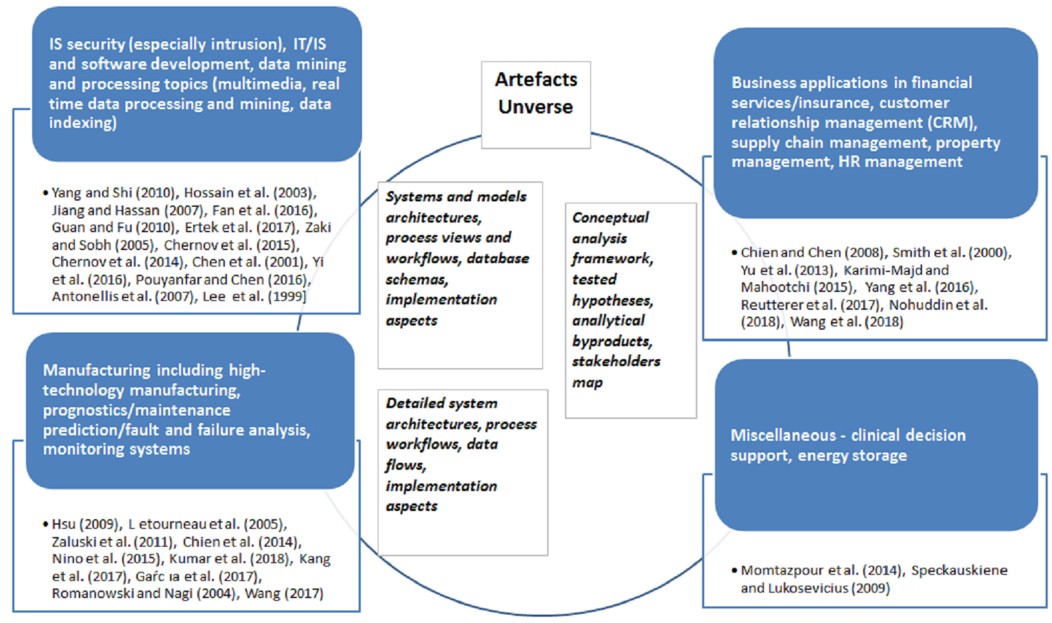

**Figure 9** 'Modification' paradigm application studies for period 1997–2018—mapping to domains.

### IT, IS domain

The largest number of publications (14 or app. 40%), was performed on IT, IS security, software development, specific data mining and processing topics. Authors address intrusion detection problem in *Hossain, Bridges & Vaughn (2003)*, *Fan, Ye & Chen (2016)*, *Lee, Stolfo & Mok (1999)*, specialized algorithms for variety of data types processing in *Yang & Shi (2010)*, *Chen et al. (2001)*, *Yi, Teng & Xu (2016)*, *Pouyanfar & Chen (2016)*, effective and efficient computer and mobile networks management in *Guan & Fu (2010)*, *Ertek, Chi & Zhang (2017)*, *Zaki & Sobh (2005)*, *Chernov, Petrov & Ristaniemi (2015)*, *Chernov et al. (2014)*.

### Manufacturing and engineering

The next most popular research area is manufacturing/engineering with 10 case studies. The central topic here is high-technology manufacturing, for example, semi-conductors associated—study of *Chien, Diaz & Lan (2014)*, and various complex prognostics case studies in rail, aerospace domains (*Létourneau et al., 2005*; *Zaluski et al., 2011*) concentrated on failure predictions. These are complemented by studies on equipment fault and failure predictions and maintenance (*Kumar, Shankar & Thakur, 2018*; *Kang et al., 2017*; *Wang, 2017*) as well as monitoring system (*García et al., 2017*).

### Sales and services, incl. financial industry

The third category is presented by seven business application papers concerning customer service, targeting and advertising (*Karimi-Majd & Mahootchi, 2015*; *Reutterer et al., 2017*; *Wang, 2017*), financial services credit risk assessments (*Smith, Willis & Brooks, 2000*), supply chain management (*Nohuddin et al., 2018*), and property management (*Yu, Fung & Haghighat, 2013*), and similar.

As a consequence of specialization, these studies concentrate on developing 'state-of-the art' solution to the respective domain-specific problem.

## Extension

'Extension' scenario was identified in 46 peer-reviewed and 12 'grey' publications. We noted that 'Extension' to existing data mining methodologies were executed with four major purposes:

1. Purpose 1: **To implement fully scaled, integrated data mining solution and regular, repeatable knowledge discovery process**—address model, algorithm deployment, implementation design (including architecture, workflows and corresponding IS integration). Also, complementary goal is to tackle changes to business process to incorporate data mining into organization activities.
2. Purpose 2: **To implement complex, specifically designed systems and integrated business applications with data mining model/solution as component or tool.** Typically, this adaptation is also oriented towards Big Data specifics, and is complemented by proposed artifacts such as Big Data architectures, system models, workflows, and data flows.
3. Purpose 3: **To implement data mining as part of integrated/combined specialized infrastructure, data environments and types (e.g., IoT, cloud, mobile networks)**.
4. Purpose 4: **To incorporate context-awareness aspects.**

The specific list of studies mapped to each of the given purposes presented in the Appendix (Table A1). Main purposes of adaptations, associated gaps and/or benefits along with observations and artifacts are documented in the Fig. 10 below.

In 'Extension' category, studies executed with the Purpose 1 propose fully scaled, integrated data mining solutions of specific data mining models, associated frameworks and processes. The distinctive trait of this research subclass is that it ensures repeatability and reproducibility of delivered data mining solution in different organizational and industry settings. Both the results of data mining use case as well as deployment and integration into IS/IT systems and associated business process(es) are presented explicitly. Thus, 'Extension' subclass is geared towards specific solution design, tackling concrete business or industrial setting problem or addressing specific research gaps thus resembling comprehensive case study.

This direction can be well exemplified by expert finder system in research social network services proposed by *Sun et al. (2015)*, data mining solution for functional test content optimization by *Wang (2015)* and time-series mining framework to conduct estimation of unobservable time-series by *Hu et al. (2010)*. Similarly, *Du et al. (2017)* tackle online log anomalies detection, automated association rule mining is addressed by *Çinicioğlu et al. (2011)*, software effort estimation by *Deng, Purvis & Purvis (2011)*, network patterns visual discovery by *Simoff & Galloway (2008)*. Number of studies address solutions in IS security (*Shin & Jeong, 2005*), manufacturing (*Güder et al., 2014*; *Chee, Baharudin & Karkonasasi, 2016*), materials engineering domains (*Doreswamy, 2008*), and business domains (*Xu & Qiu, 2008*; *Ding & Daniel, 2007*).

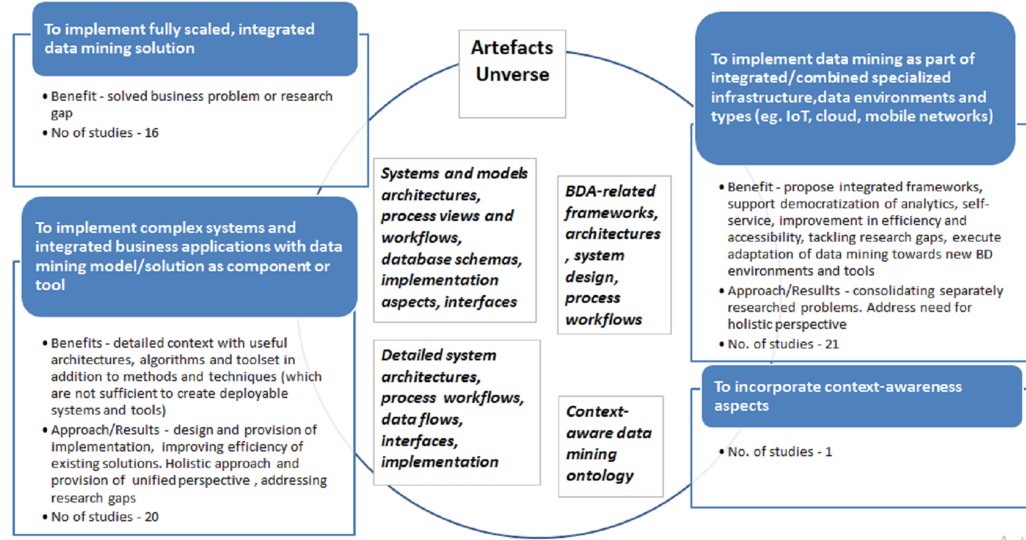

**Figure 10** 'Extension' scenario adaptations goals, benefits, artifacts and number of publications for period 1997–2018.

In contrast, 'Extension' studies executed for the Purpose 2 concentrate on design of complex, multi-component information systems and architectures. These are holistic, complex systems and integrated business applications with data mining framework serving as component or tool. Moreover, data mining methodology in these studies is extended with systems integration phases.

For example, *Mobasher (2007)* presents data mining application in Web personalization system and associated process; here, data mining cycle is extended in all phases with utmost goal of leveraging multiple data sources and using discovered models and corresponding algorithms in an automatic personalization system. Authors comprehensively address data processing, algorithm, design adjustments and respective integration into automated system. Similarly, *Haruechaiyasak, Shyu & Chen (2004)* tackle improvement of Webpage recommender system by presenting extended data mining methodology including design and implementation of data mining model. Holistic view on web-mining with support of all data sources, data warehousing and data mining techniques integration, as well as multiple problem-oriented analytical outcomes with rich business application scenarios (personalization, adaptation, profiling, and recommendations) in e-commerce domain was proposed and discussed by *Büchner & Mulvenna (1998)*. Further, *Singh et al. (2014)* tackled scalable implementation of Network Threat Intrusion Detection System. In this study, data mining methodology and resulting model are extended, scaled and deployed as module of quasi-real-time system for capturing Peer-to-Peer Botnet attacks. Similar complex solution was presented in a series of publications by *Lee et al. (2000, 2001)* who designed real-time data mining-based Intrusion Detection System (IDS). These works are complemented by comprehensive study of *Barbará et al. (2001)* who constructed experimental testbed for intrusion detection with data mining methods. Detection model combining data fusion and mining and respective components for Botnets identification was developed by *Kiayias et al. (2009)* too. Similar approach is presented in *Alazab et al. (2011)* who proposed

and implemented zero-day malware detection system with associated machine-learning based framework. Finally, *Ahmed, Rafique & Abulaish (2011)* presented multi-layer framework for fuzzy attack in 3G cellular IP networks.

A number of authors have considered data mining methodologies in the context of Decision Support Systems and other systems that generate information for decision-making, across a variety of domains. For example, *Kisilevich, Keim & Rokach (2013)* executed significant extension of data mining methodology by designing and presenting integrated Decision Support System (DSS) with six components acting as supporting tool for hotel brokerage business to increase deal profitability. Similar approach is undertaken by *Capozzoli et al. (2017)* focusing on improving energy management of properties by provision of occupancy pattern information and reconfiguration framework. *Kabir (2016)* presented data mining information service providing improved sales forecasting that supported solution of under/over-stocking problem while *Lau, Zhang & Xu (2018)* addressed sales forecasting with sentiment analysis on Big Data. *Kamrani, Rong & Gonzalez (2001)* proposed GA-based Intelligent Diagnosis system for fault diagnostics in manufacturing domain. The latter was tackled further in *Shahbaz et al. (2010)* with complex, integrated data mining system for diagnosing and solving manufacturing problems in real time.

*Lenz, Wuest & Westkämper (2018)* propose a framework for capturing data analytics objectives and creating holistic, cross-departmental data mining systems in the manufacturing domain. This work is representative of a cohort of studies that aim at extending data mining methodologies in order to support the design and implementation of enterprise-wide data mining systems. In this same research cohort, we classify *Luna, Castro & Romero (2017)*, which presents a data mining toolset integrated into the Moodle learning management system, with the aim of supporting university-wide learning analytics.

One study addresses multi-agent based data mining concept. *Khan, Mohamudally & Babajee (2013)* have developed unified theoretical framework for data mining by formulating a unified data mining theory. The framework is tested by means of agent programing proposing integration into multi-agent system which is useful due to scalability, robustness and simplicity.

The subcategory of 'Extension' research executed with Purpose 3 is devoted to data mining methodologies and solutions in specialized IT/IS, data and process environments which emerged recently as consequence of Big Data associated technologies and tools development. Exemplary studies include IoT associated environment research, for example, Smart City application in IoT presented by *Strohbach et al. (2015)*. In the same domain, *Bashir & Gill (2016)* addressed IoT-enabled smart buildings with the additional challenge of large amount of high-speed real time data and requirements of real-time analytics. Authors proposed integrated IoT Big Data Analytics framework. This research is complemented by interdisciplinary study of *Zhong et al. (2017)* where IoT and wireless technologies are used to create RFID-enabled environment producing analysis of KPIs to improve logistics.

Significant number of studies addresses various mobile environments sometimes complemented by cloud-based environments or cloud-based environments as stand-alone.

*Gomes, Phua & Krishnaswamy (2013)* addressed mobile data mining with execution on mobile device itself; the framework proposes innovative approach addressing extensions of all aspects of data mining including contextual data, end-user privacy preservation, data management and scalability. *Yuan, Herbert & Emamian (2014)* and *Yuan & Herbert (2014)* introduced cloud-based mobile data analytics framework with application case study for smart home based monitoring system. *Cuzzocrea, Psaila & Toccu (2016)* have presented innovative FollowMe suite which implements data mining framework for mobile social media analytics with several tools with respective architecture and functionalities. An interesting paper was presented by *Torres et al. (2017)* who addressed data mining methodology and its implementation for congestion prediction in mobile LTE networks tackling also feedback reaction with network reconfigurations trigger.

Further, *Biliri et al. (2014)* presented cloud-based Future Internet Enabler—automated social data analytics solution which also addresses Social Network Interoperability aspect supporting enterprises to interconnect and utilize social networks for collaboration. Real-time social media streamed data and resulting data mining methodology and application was extensively discussed by *Zhang, Lau & Li (2014)*. Authors proposed design of comprehensive ABIGDAD framework with seven main components implementing data mining based deceptive review identification. Interdisciplinary study tackling both these topics was developed by *Puthal et al. (2016)* who proposed integrated framework and architecture of disaster management system based on streamed data in cloud environment ensuring end-to-end security. Additionally, key extensions to data mining framework have been proposed merging variety of data sources and types, security verification and data flow access controls. Finally, cloud-based manufacturing was addressed in the context of fault diagnostics by *Kumar et al. (2016)*.

Also, *Mahmood et al. (2013)* tackled Wireless Sensor Networks and associated data mining framework required extensions. Interesting work is executed by *Nestorov & Jukic (2003)* addressing rare topic of data mining solutions integration within traditional data warehouses and active mining of data repositories themselves.

Supported by new generation of visualization technologies (including Virtual Reality environments), *Wijayasekara, Linda & Manic (2011)* proposed and implemented CAVE-SOM (3D visual data mining framework) which offers interactive, immersive visual data mining with multiple visualization modes supported by plethora of methods. Earlier version of visual data mining framework was successfully developed and presented by *Ganesh et al. (1996)* as early as in 1996.

Large-scale social media data is successfully tackled by *Lemieux (2016)* with comprehensive framework accompanied by set of data mining tools and interface. Real time data analytics was addressed by *Shrivastava & Pal (2017)* in the domain of enterprise service ecosystem. Images data was addressed in *Huang et al. (2002)* by proposing multimedia data mining framework and its implementation with user relevance feedback integration and instance learning. Further, exploded data diversity and associated need to extend standard data mining is addressed by *Singh et al. (2016)* in the study devoted to object detection in video surveillance systems supporting real time video analysis.

Finally, there is also limited number of studies which addresses context awareness (Purpose 4) and extends data mining methodology with context elements and adjustments. In comparison with 'Integration' category research, here, the studies are at lower abstraction level, capturing and presenting list of adjustments. *Singh, Vajirkar & Lee (2003)* generate taxonomy of context factors, develop extended data mining framework and propose deployment including detailed IS architecture. Context-awareness aspect is also addressed in the papers reviewed above, for example, *Lenz, Wuest & Westkämper (2018)*, *Kisilevich, Keim & Rokach (2013)*, *Sun et al. (2015)*, and other studies.

## Integration

'Integration' of data mining methodologies scenario was identified in 27 'peer-reviewed' and 17 'grey' studies. Our analysis revealed that this adaptation scenario at a higher abstraction level is typically executed with the five key purposes:

1. Purpose 1: **to integrate/combine with various ontologies existing in organization**.
2. Purpose 2: **to introduce context-awareness and incorporate domain knowledge**.
3. Purpose 3: **to integrate/combine with other research or industry domains framework, process methodologies and concepts**.
4. Purpose 4: **to integrate/combine with other well-known organizational governance frameworks, process methodologies and concepts**.
5. Purpose 5: **to accommodate and/or leverage upon newly available Big Data technologies, tools and methods.**

The specific list of studies mapped to each of the given purposes presented in Appendix (Table A2). Main purposes of adaptations, associated gaps and/or benefits along with observations and artifacts are documented in Fig. 11 below.

As mentioned, number of studies concentrates on proposing ontology-based Integrated data mining frameworks accompanies by various types of ontologies (Purpose 1). For example, *Sharma & Osei-Bryson (2008)* focus on ontology-based organizational view with Actors, Goals and Objectives which supports execution of Business Understanding Phase. *Brisson & Collard (2008)* propose KEOPS framework which is CRISP-DM compliant and integrates a knowledge base and ontology with the purpose to build ontology-driven information system (OIS) for business and data understanding phases while knowledge base is used for post-processing step of model interpretation. *Park et al. (2017)* propose and design comprehensive ontology-based data analytics tool IRIS with the purpose to align analytics and business. IRIS is based on concept to connect dots, analytics methods or transforming insights into business value, and supports standardized process for applying ontology to match business problems and solutions.

Further, *Ying et al. (2014)* propose domain-specific data mining framework oriented to business problem of customer demand discovery. They construct ontology for customer demand and customer demand discovery task which allows to execute structured knowledge extraction in the form of knowledge patterns and rules. Here, the purpose is to facilitate business value realization and support actionability of extracted knowledge

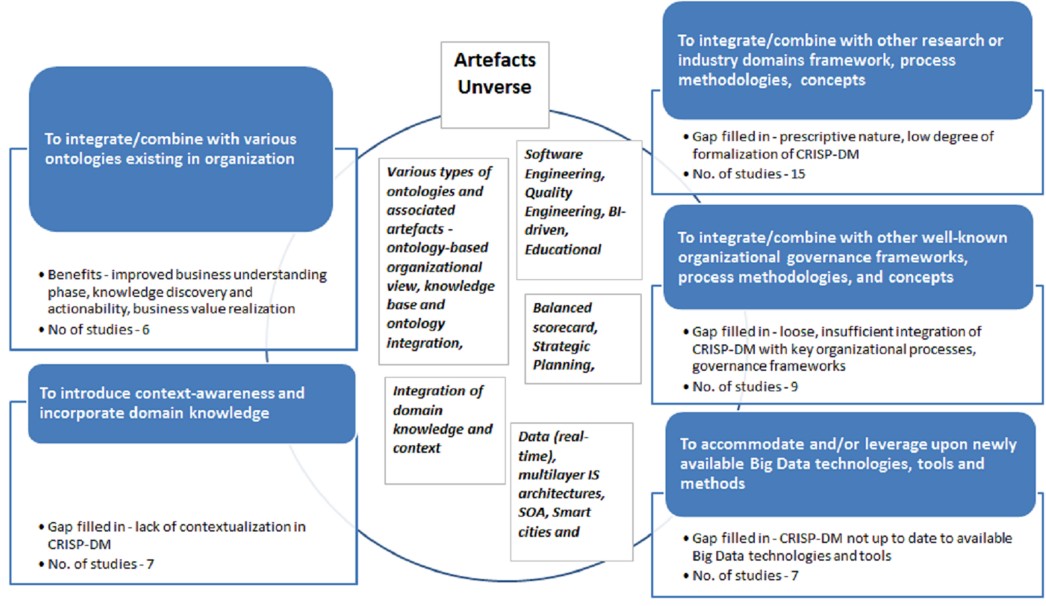

**Figure 11 'Integration' scenario adaptations goals, benefits, artifacts and number of publications for period 1997–2018.**

via marketing strategies and tactics. In the same vein, *Cannataro & Comito (2003)* presented ontology for the Data Mining domain which main goal is to simplify the development of distributed knowledge discovery applications. Authors offered to a domain expert a reference model for different kind of data mining tasks, methodologies, and software capable to solve the given business problem and find the most appropriate solution.

Apart from ontologies, *Sharma & Osei-Bryson (2009)* in another study propose IS inspired, driven by Input-Output model data mining methodology which supports formal implementation of Business Understanding Phase. This research exemplifies studies executed with Purpose 2. The goal of the paper is to tackle prescriptive nature of CRISP-DM and address how the entire process can be implemented. *Cao, Schurmann & Zhang (2005)* study is also exemplary in terms of aggregating and introducing several fundamental concepts into traditional CRISP-DM data mining cycle—context awareness, in-depth pattern mining, human–machine cooperative knowledge discovery (in essence, following human-centricity paradigm in data mining), loop-closed iterative refinement process (similar to Agile-based methodologies in Software Development). There are also several concepts, like data, domain, interestingness, rules which are proposed to tackle number of fundamental constrains identified in CRISP-DM. They have been discussed and further extended by *Cao & Zhang (2007, 2008)*, *Cao (2010)* into integrated domain driven data mining concept resulting in fully fledged D3M (domain-driven) data mining framework. Interestingly, the same concepts, but on individual basis are investigated and presented by other authors, for example, context-aware data mining methodology is tackled by *Xiang (2009a, 2009b)* in the context of financial sector. *Pournaras et al. (2016)* attempted very crucial privacy-preservation topic in the context of achieving effective data analytics

methodology. Authors introduced metrics and self-regulatory (reconfigurable) information sharing mechanism providing customers with controls for information disclosure.

A number of studies have proposed CRISP-DM adjustments based on existing frameworks, process models or concepts originating in other domains (Purpose 3), for example, software engineering (*Marbán et al., 2007, 2009; Marban, Mariscal & Segovia, 2009*) and industrial engineering (*Solarte, 2002; Zhao et al., 2005*).

Meanwhile, *Mariscal, Marbán & Fernández (2010)* proposed a new refined data mining process based on a global comparative analysis of existing frameworks while *Angelov (2014)* outlined a data analytics framework based on statistical concepts. Following a similar approach, some researchers suggest explicit integration with other areas and organizational functions, for example, BI-driven Data Mining by *Hang & Fong (2009)*. Similarly, *Chen, Kazman & Haziyev (2016)* developed an architecture-centric agile Big Data analytics methodology, and an architecture-centric agile analytics and DevOps model. Alternatively, several authors tackled data mining methodology adaptations in other domains, for example, educational data mining by *Tavares, Vieira & Pedro (2017)*, decision support in learning management systems (*Murnion & Helfert, 2011*), and in accounting systems (*Amani & Fadlalla, 2017*).

Other studies are concerned with actionability of data mining and closer integration with business processes and organizational management frameworks (Purpose 4). In particular, there is a recurrent focus on embedding data mining solutions into knowledge-based decision making processes in organizations, and supporting fast and effective knowledge discovery (*Bohanec, Robnik-Sikonja & Borstnar, 2017*).

Examples of adaptations made for this purpose include: (1) integration of CRISP-DM with the Balanced Scorecard framework used for strategic performance management in organizations (*Yun, Weihua & Yang, 2014*); (2) integration with a strategic decision-making framework for revenue management *Segarra et al. (2016)*; (3) integration with a strategic analytics methodology *Van Rooyen & Simoff (2008)*, and (4) integration with a so-called 'Analytics Canvas' for management of portfolios of data analytics projects *Kühn et al. (2018)*. Finally, *Ahangama & Poo (2015)* explored methodological attributes important for adoption of data mining methodology by novice users. This latter study uncovered factors that could support the reduction of resistance to the use of data mining methodologies. Conversely, *Lawler & Joseph (2017)* comprehensively evaluated factors that may increase the benefits of Big Data Analytics projects in an organization.

Lastly, a number of studies have proposed data mining frameworks (e.g., CRISP-DM) adaptations to cater for new technological architectures, new types of datasets and applications (Purpose 5). For example, *Lu et al. (2017)* proposed a data mining system based on a Service-Oriented Architecture (SOA), *Zaghloul, Ali-Eldin & Salem (2013)* developed a concept of self-service data analytics, *Osman, Elragal & Bergvall-Kåreborn (2017)* blended CRISP-DM into a Big Data Analytics framework for Smart Cities, and *Niesen et al. (2016)* proposed a data-driven risk management framework for Industry 4.0 applications.

Our analysis of RQ3, regarding the purposes of existing data mining methodologies adaptations, revealed the following key findings. Firstly, adaptations of type 'Modification' are predominantly targeted at addressing problems that are specific to a given case study.

The majority of modifications were made within the domain of IS security, followed by case studies in the domains of manufacturing and financial services. This is in clear contrast with adaptations of type 'Extension', which are primarily aimed at customizing the methodology to take into account specialized development environments and deployment infrastructures, and to incorporate context-awareness aspects. Thirdly, a recurrent purpose of adaptations of type 'Integration' is to combine a data mining methodology with either existing ontologies in an organization or with other domain frameworks, methodologies, and concepts. 'Integration' is also used to instill context-awareness and domain knowledge into a data mining methodology, or to adapt it to specialized methods and tools, such as Big Data. The distinctive outcome and value (gaps filled in) of 'Integrations' stems from improved knowledge discovery, better actionability of results, improved combination with key organizational processes and domain-specific methodologies, and improved usage of Big Data technologies.

### Summary

We discovered that the adaptations of existing data mining methodologies found in the literature can be classified into three categories: modification, extension, or integration.

We also noted that adaptations are executed either to address deficiencies and lack of important elements or aspects in the reference methodology (chiefly CRISP-DM). Furthermore, adaptations are also made to improve certain phases, deliverables or process outcomes.

In short, adaptations are made to:

- improve key reference data mining methodologies phases—for example, in case of CRISP-DM these are primarily business understanding and deployment phases.
- support knowledge discovery and actionability.
- introduce context-awareness and higher degree of formalization.
- integrate closer data mining solution with key organizational processes and frameworks.
- significantly update CRISP-DM with respect to Big Data technologies, tools, environments and infrastructure.
- incorporate broader, explicit context of architectures, algorithms and toolsets as integral deliverables or supporting tools to execute data mining process.
- expand and accommodate broader unified perspective for incorporating and implementing data mining solutions in organization, IT infrastructure and business processes.

## THREATS TO VALIDITY

Systematic literature reviews have inherent limitations that must be acknowledged. These threats to validity include subjective bias (internal validity) and incompleteness of search results (external validity).

The internal validity threat stems from the subjective screening and rating of studies, particularly when assessing the studies with respect to relevance and quality criteria.

We have mitigated these effects by documenting the survey protocol (SLR Protocol), strictly adhering to the inclusion criteria, and performing significant validation procedures, as documented in the Protocol.

The external validity threat relates to the extent to which the findings of the SLR reflect the actual state of the art in the field of data mining methodologies, given that the SLR only considers published studies that can be retrieved using specific search strings and databases. We have addressed this threat to validity by conducting trial searches to validate our search strings in terms of their ability to identify relevant papers that we knew about beforehand. Also, the fact that the searches led to 1,700 hits overall suggests that a significant portion of the relevant literature has been covered.

## CONCLUSION

In this study, we have examined the use of data mining methodologies by means of a systematic literature review covering both peer-reviewed and 'grey' literature. We have found that the use of data mining methodologies, as reported in the literature, has grown substantially since 2007 (four-fold increase relative to the previous decade). Also, we have observed that data mining methodologies were predominantly applied 'as-is' from 1997 to 2007. This trend was reversed from 2008 onward, when the use of adapted data mining methodologies gradually started to replace 'as-is' usage.

The most frequent adaptations have been in the 'Extension' category. This category refers to adaptations that imply significant changes to key phases of the reference methodology (chiefly CRISP-DM). These adaptations particularly target the business understanding, deployment and implementation phases of CRISP-DM (or other methodologies). Moreover, we have found that the most frequent purposes of adaptions are: (1) adaptations to handle Big Data technologies, tools and environments (technological adaptations); and (2) adaptations for context-awareness and for integrating data mining solutions into business processes and IT systems (organizational adaptations). A key finding is that standard data mining methodologies do not pay sufficient attention to deployment aspects required to scale and transform data mining models into software products integrated into large IT/IS systems and business processes.

Apart from the adaptations in the 'Extension' category, we have also identified an increasing number of studies focusing on the 'Integration' of data mining methodologies with other domain-specific and organizational methodologies, frameworks, and concepts. These adaptions are aimed at embedding the data mining methodology into broader organizational aspects.

Overall, the findings of the study highlight the need to develop refinements of existing data mining methodologies that would allow them to seamlessly interact with IT development platforms and processes (technological adaptation) and with organizational management frameworks (organizational adaptation). In other words, there is a need to frame existing data mining methodologies as being part of a broader ecosystem of methodologies, as opposed to the traditional view where data mining methodologies are defined in isolation from broader IT systems engineering and organizational management methodologies.

# APPENDICES

| Table A1 'Extension' paradigm data mining methodologies application studies for period 1997–2018. | |
|---|---|
| **Main adaptation purpose** | **Publications** |
| (1) To implement fully scaled, integrated data mining solution | *Sun et al. (2015)*, *Hu et al. (2010)*, *Wang (2015)*, *Du et al. (2017)*, *Çinicioğlu et al. (2011)*, *Doreswamy (2008)*, *Güder et al. (2014)*, *Simoff & Galloway (2008)*, *Deng, Purvis & Purvis (2011)*, *Xu & Qiu (2008)*, *Shin & Jeong (2005)*, *Chee, Baharudin & Karkonasasi (2016)*, *Zhang (2009)*, *Ding & Daniel (2007)*, *Liu et al. (2018)*, *Shao, Liu & Zhu (2008)* |
| (2) To implement complex systems and integrated business applications with data mining model/solution as component or tool | *Mobasher (2007)*, *Singh et al. (2014)*, *Alazab et al. (2011)*, *Kisilevich, Keim & Rokach (2013)*, *Haruechaiyasak, Shyu & Chen (2004)*, *Luna, Castro & Romero (2017)*, *Khan, Mohamudally & Babajee (2013)*, *Ortega et al. (2015)*, *Lau, Zhang & Xu (2018)*, *Ahmed, Rafique & Abulaish (2011)*, *Capozzoli et al. (2017)*, *Kabir (2016)*, *Kiayias et al. (2009)*, *Kamrani, Rong & Gonzalez (2001)*, *Büchner & Mulvenna (1998)*, *Shahbaz et al. (2010)*, *Lee et al. (2000, 2001)*, *Barbará et al. (2001)*, *Lenz, Wuest & Westkämper (2018)* |
| (3) To implement data mining as part of integrated/combined specialized infrastructure,data environments and types (e.g., IoT, cloud, mobile networks) | *Strohbach et al. (2015)*, *Mahmood et al. (2013)*, *Nestorov & Jukic (2003)*, *Gomes, Phua & Krishnaswamy (2013)*, *Wijayasekara, Linda & Manic (2011)*, *Yuan & Herbert (2014)*, *Bashir & Gill (2016)*, *Cuzzocrea, Psaila & Toccu (2016)*, *Biliri et al. (2014)*, *Rendall et al. (2017)*, *Zhang, Lau & Li (2014)*, *Yuan, Herbert & Emamian (2014)*, *Huang et al. (2002)*, *Singh et al. (2016)*, *Shrivastava & Pal (2017)*, *Lemieux (2016)*, *Ganesh et al. (1996)*, *Torres et al. (2017)*, *Zhong et al. (2017)*, *Puthal et al. (2016)*, *Kumar et al. (2016)* |
| (4) To incorporate context-awareness aspects | *Singh, Vajirkar & Lee (2003)* |

| Table A2 'Integration' paradigm data mining methodologies application studies for period 1997–2018. | |
|---|---|
| **Main adaptation purpose** | **Publications** |
| (1) To integrate/combined with various ontologies existing in organization | *Sharma & Osei-Bryson (2008, 2009)*, *Brisson & Collard (2008)*, *Park et al. (2017)*, *Ying et al. (2014)*, *Cannataro & Comito (2003)* |
| (2) To introduce context-awareness and incorporate domain knowledge | *Cao, Schurmann & Zhang (2005)*, *Cao & Zhang (2007, 2008)*, *Xiang (2009a, 2009b)*, *Pournaras et al. (2016)*, *Cao (2010)* |
| (3) To integrate/combine with other research/industry domains frameworks, process methodologies, and concepts | *Marbán et al. (2007)*, *Zhao et al. (2005)*, *François (2008)*, *Hang & Fong (2009)*, *Tavares, Vieira & Pedro (2017)*, *Murnion & Helfert (2011)*, *Amani & Fadlalla (2017)*, *Marban, Mariscal & Segovia (2009)*, *Mariscal, Marbán & Fernández (2010)*, *Solarte (2002)*, *Marbán et al. (2009)*, *Chen, Kazman & Haziyev (2016)*, *Ahangama & Poo (2015)*, *Angelov (2014)* |
| (4) To integrate/combine with other organizational governance frameworks, process methodologies, concepts | *Bohanec, Robnik-Sikonja & Borstnar (2017)*, *Debuse (2007)*, *Chatzikonstantinou, Kontogiannis & Attarian (2013)*, *Rahman, Desa & Wibowo (2011)*, *Yun, Weihua & Yang (2014)*, *Van Rooyen & Simoff (2008)*, *Kühn et al. (2018)*, *Segarra et al. (2016)*, *Lawler & Joseph (2017)* |
| (5) To accomodate or leverage upon newly available Big Data technologies, tools and methods | *Lu et al. (2017)*, *Osman, Elragal & Bergvall-Kåreborn (2017)*, *Behbahani, Khaddaj & Choudhury (2011)*, *Deng, Ghanem & Guo (2009)*, *Kurgan & Musilek (2006)*, *Zaghloul, Ali-Eldin & Salem (2013)*, *Niesen et al. (2016)* |

### Funding

The authors received no funding for this work.

### Competing Interests

The authors declare that they have no competing interests.

## Author Contributions

- Veronika Plotnikova conceived and designed the experiments, performed the experiments, analyzed the data, performed the computation work, prepared figures and/ or tables, authored or reviewed drafts of the paper, and approved the final draft.
- Marlon Dumas conceived and designed the experiments, authored or reviewed drafts of the paper, and approved the final draft.
- Fredrik Milani conceived and designed the experiments, authored or reviewed drafts of the paper, and approved the final draft.

## Data Availability

SLR Protocol (also shared via online repository), corpus with definitions and mappings are provided as a Supplemental File.

## Supplemental Information

Supplemental information for this article can be found online at http://dx.doi.org/10.7717/peerj-cs.267#supplemental-information.

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

## PRIMARY SOURCES

**Ahangama S, Poo DCC. 2015.** What methodological attributes are essential for novice users to analytics? An empirical study. In: *17th International Conference on HCI International 2015 Human Interface and the Management of Information, Information and Knowledge in Context, 2–7 August 2015, Los Angeles, CA, USA.* 77–88.

**Ahmed F, Rafique MZ, Abulaish M. 2011.** A data mining framework for securing 3G core network from GTP fuzzing attacks. In: *7th International Conference on ICISS 2011 Information Systems Security, 15–19 December 2011, Kolkata, India.* 280–293.

**Alazab M, Venkatraman S, Watters PA, Alazab M. 2011.** Zero-day malware detection based on supervised learning algorithms of API call signatures. In: *Ninth Australasian Data Mining Conference, AusDM 2011, Ballarat, Australia.* 171–182.

**Amani F, Fadlalla A. 2017.** Data mining applications in accounting: a review of the literature and organizing framework. *International Journal of Accounting Information Systems* **24**:32–58 DOI 10.1016/j.accinf.2016.12.004.

**Angelov P. 2014.** Outside the box: an alternative data analytics framework. *Journal of Automation Mobile Robotics and Intelligent Systems* **8(2)**:29–35 DOI 10.14313/JAMRIS_2-2014/16.

**Barbará D, Couto J, Jajodia S, Wu N. 2001.** ADAM: a testbed for exploring the use of data mining in intrusion detection. *SIGMOD Record* **30(4)**:15–24 DOI 10.1145/604264.604268.

**Bashir MR, Gill AQ. 2016.** Towards an IOT big data analytics framework: smart buildings systems. In: *18th IEEE International Conference on High Performance Computing and Communications; 14th IEEE International Conference on Smart City; 2nd IEEE International Conference on Data Science and Systems, HPCC/SmartCity/DSS 2016, 12–14 December 2016, Sydney, Australia.* 1325–1332.

**Behbahani MP, Khaddaj S, Choudhury I. 2011.** A multilayer data mining approach to an optimized ebusiness analytics framework. In: *International Proceedings of Economics Development and Research*, 66–71.

**Biliri E, Petychakis M, Alvertis I, Lampathaki F, Koussouris S, Askounis D. 2014.** Infusing social data analytics into future internet applications for manufacturing. In: *11th IEEE/ACS International Conference on Computer Systems and Applications, AICCSA 2014, 10–13 November 2014, Doha, Qatar.* 515–522.

**Bohanec M, Robnik-Sikonja M, Borstnar MK. 2017.** Decision-making framework with double-loop learning through interpretable black-box machine learning models. *Industrial Management and Data Systems* **117(7)**:1389–1406 DOI 10.1108/IMDS-09-2016-0409.

**Brisson L, Collard M. 2008.** How to semantically enhance a data mining process? In: *10th International Conference on Enterprise Information Systems, ICEIS 2008, 12–16 June 2008, Barcelona, Spain.* 103–116.

**Büchner AG, Mulvenna MD. 1998.** Discovering internet marketing intelligence through online analytical web usage mining. *SIGMOD Record* **27(4)**:54–61.

**Cannataro M, Comito C. 2003.** A data mining ontology for grid programming. In: *Proceedings of the 1st International Workshop on Semantics in Peer-to-Peer and Grid Computing, Citeseer.* 113–134.

**Cao L. 2010.** Domain-driven data mining: challenges and prospects. *IEEE Transactions on Knowledge and Data Engineering* **22(6)**:755–769 DOI 10.1109/TKDE.2010.32.

**Cao L, Schurmann R, Zhang C. 2005.** Domain-driven in-depth pattern discovery: a practical methodology. In: *Australian Data Mining Conference*, Sydney: The University of Technology.

**Cao L, Zhang C. 2007.** The evolution of KDD: towards domain-driven data mining. *IJPRAI* **21(4)**:677–692.

**Cao L, Zhang C. 2008.** Domain driven data mining. In: *Data Mining and Knowledge Discovery Technologies*, IGI Global, 196–223.

**Capozzoli A, Piscitelli MS, Gorrino A, Ballarini I, Corrado V. 2017.** Data analytics for occupancy pattern learning to reduce the energy consumption of HVAC systems in office buildings. *Sustainable Cities and Society* **35**:191–208 DOI 10.1016/j.scs.2017.07.016.

**Chatzikonstantinou G, Kontogiannis K, Attarian I. 2013.** A goal driven framework for software project data analytics. In: *25th International Conference Advanced Information Systems Engineering, CAiSE 2013, 17–21 June 2013, Valencia, Spain.* 546–561.

**Chee TC, Baharudin AS, Karkonasasi K. 2016.** Data mining framework for test time optimization in industrial electronics manufacturing enterprise. *International Journal of Applied Engineering Research* **11(4)**:2717–2722.

**Chen H, Kazman R, Haziyev S. 2016.** Agile big data analytics development: an architecture-centric approach. In: *49th Hawaii International Conference on System Sciences, HICSS 2016, 5–8 January 2016, Koloa, HI, USA.* 5378–5387.

**Chen Y, Gao W, Wang Z, Miao J, Jiang D. 2001.** Mining audio/visual database for speech driven face animation. In: *Proceedings of the IEEE International Conference on Systems, Man & Cybernetics: e-Systems and e-Man for Cybernetics in Cyberspace, 7–10 October 2001, Tucson, AZ, USA.* 2638–2643.

**Chernov S, Chernogorov F, Petrov D, Ristaniemi T. 2014.** Data mining framework for random access failure detection in LTE networks. In: *25th IEEE Annual International Symposium on Personal, Indoor, and Mobile Radio Communication, PIMRC 2014, 2–5 September 2014, Washington DC, USA.* 1321–1326.

**Chernov S, Petrov D, Ristaniemi T. 2015.** Location accuracy impact on cell outage detection in LTE-A networks. In: *International Wireless Communications and Mobile Computing Conference, IWCMC 2015, 24–28 August 2015, Dubrovnik, Croatia.* 1162–1167.

**Chien C, Diaz AC, Lan Y. 2014.** A data mining approach for analyzing semiconductor MES and FDC data to enhance overall usage effectiveness (OUE). *International Journal of Computational Intelligence Systems* **7(Suppl. 2)**:52–65 DOI 10.1080/18756891.2014.947114.

**Çinicioğlu EN, Ertek G, Demirer D, Yörük HE. 2011.** A framework for automated association mining over multiple databases. In: *2011 International Symposium on Innovations in Intelligent Systems and Applications.* IEEE, 79–85.

**Cuzzocrea A, Psaila G, Toccu M. 2016.** An innovative framework for effectively and efficiently supporting big data analytics over geo-located mobile social media. In: *Proceedings of the 20th International Database Engineering & Applications Symposium, IDEAS 2016, 11–13 July 2016, Montreal, QC, Canada.* 62–69.

**Debuse J. 2007.** Extending data mining methodologies to encompass organizational factors. *Systems Research and Behavioral Science: Official Journal of the International Federation for Systems Research* **24(2)**:183–190 DOI 10.1002/sres.823.

**Deng JD, Purvis MK, Purvis M. 2011.** Software effort estimation: harmonizing algorithms and domain knowledge in an integrated data mining approach. *IJIIT* **7(3)**:41–53.

**Deng X, Ghanem M, Guo Y. 2009.** Real-time data mining methodology and a supporting framework. In: *Third International Conference on Network and System Security, NSS 2009, 19–21 October 2009, Gold Coast, QLD, Australia.* 522–527.

**Ding Q, Daniel C. 2007.** Multimedia data mining framework for banner images. In: Petrushin VA, Khan L, eds. *Multimedia Data Mining and Knowledge Discovery*. London: Springer, 448–457.

**Doreswamy H. 2008.** A survey for data mining frame work for polymer matrix composite engineering materials design applications. *International Journal of Computational Intelligence Systems* **1(4)**:313–328.

**Du M, Li F, Zheng G, Srikumar V. 2017.** Deeplog: Anomaly detection and diagnosis from system logs through deep learning. In: *Proceedings of the 2017 ACM SIGSAC Conference on Computer and Communications Security, CCS 2017, 30 October–3 November 2017, Dallas, TX, USA.* 1285–1298.

**Ertek G, Chi X, Zhang AN. 2017.** A framework for mining RFID data from schedule-based systems. *IEEE Transactions on Systems, Man, and Cybernetics: Systems* **47(11)**:2967–2984 DOI 10.1109/TSMC.2016.2557762.

**Fan Y, Ye Y, Chen L. 2016.** Malicious sequential pattern mining for automatic malware detection. *Expert Systems with Applications* **52**:16–25 DOI 10.1016/j.eswa.2016.01.002.

**François D. 2008.** Methodology and standards for data analysis with machine learning tools. In: *16th European Symposium on Artificial Neural Networks, ESANN 2008, 23–25 April 2008, Bruges, Belgium.* 239–246.

**Ganesh M, Han E, Kumar V, Shekhar S, Srivastava J. 1996.** *Visual data mining: framework and algorithm development.* Minnesota: Department of Computing and Information Sciences, University of Minnesota.

**García D, Creus R, Minoves M, Pardo X, Quevedo J, Puig V. 2017.** Data analytics methodology for monitoring quality sensors and events in the Barcelona drinking water network. *Journal of Hydroinformatics* **19(1)**:123–137 DOI 10.2166/hydro.2016.048.

**Gomes JB, Phua C, Krishnaswamy S. 2013.** Where will you go? mobile data mining for next place prediction. In: *15th International Conference on Data Warehousing and Knowledge Discovery, DaWaK 2013, 26–29 August 2013, Prague, Czech Republic.* 146–158.

**Guan Q, Fu S. 2010.** Auto-aid: a data mining framework for autonomic anomaly identification in networked computer systems. In: *29th International Performance Computing and Communications Conference, IPCCC 2010, 9–11 December 2010, Albuquerque, NM, USA.* 73–80.

**Güder M, Salor-Durna Ö, Çadirci I, Ozkan B, Altintas E. 2014.** Data mining framework for power quality event characterization of iron and steel plants. In: *2014 IEEE Industry Application Society Annual Meeting, 5–9 October 2014, Vancouver, BC, Canada.* 1–11.

**Hang Y, Fong S. 2009.** A framework of business intelligence-driven data mining for e-business. In: *International Conference on Networked Computing and Advanced Information Management, NCM 2009; Fifth International Joint Conference on INC, IMS and IDC: INC 2009; International Conference on Networked Computing, IMS 2009; International Conference on Advanced Information Management and Service, IDC 2009; International Conference on Digital Content, Multimedia Technology and its Applications, 25–27 August 2009, Seoul, South Korea.* 1964–1970.

**Haruechaiyasak C, Shyu M, Chen S. 2004.** A data mining framework for building a web-page recommender system. In: *Proceedings of the 2004 IEEE International Conference on Information Reuse and Integration, IRI-2004, 8–10 November 2004, Las Vegas, NV, USA.* 357–362.

**Hossain M, Bridges SM, Vaughn RB Jr. 2003.** Adaptive intrusion detection with data mining. In: *Proceedings of the IEEE International Conference on Systems, Man & Cybernetics, 5–8 October 2003, Washington, DC, USA.* 3097–3103.

**Hu X, Xu P, Wu S, Asgari S, Bergsneider M. 2010.** A data mining framework for time series estimation. *Journal of Biomedical Informatics* **43(2)**:190–199 DOI 10.1016/j.jbi.2009.11.002.

**Huang X, Chen S, Shyu M, Zhang C. 2002.** Mining high-level user concepts with multiple instance learning and relevance feedback for content-based image retrieval. In: *Mining Multimedia and*

*Complex Data, KDD Workshop MDM/KDD 2002, PAKDD Workshop KDMCD 2002, Revised Papers*, 50–67.

**Kabir MH. 2016.** Data mining framework for generating sales decision making information using association rules. *International Journal of Advanced Computer Science and Applications* **7(5)**:378–385.

**Kamrani A, Rong W, Gonzalez R. 2001.** A genetic algorithm methodology for data mining and intelligent knowledge acquisition. *Computers & Industrial Engineering* **40(4)**:361–377 DOI 10.1016/S0360-8352(01)00036-5.

**Kang S, Kim E, Shim J, Cho S, Chang W, Kim J. 2017.** Mining the relationship between production and customer service data for failure analysis of industrial products. *Computers & Industrial Engineering* **106**:137–146 DOI 10.1016/j.cie.2017.01.028.

**Karimi-Majd A, Mahootchi M. 2015.** A new data mining methodology for generating new service ideas. *Information Systems and e-Business Management* **13(3)**:421–443 DOI 10.1007/s10257-014-0267-y.

**Khan DM, Mohamudally N, Babajee DKR. 2013.** A unified theoretical framework for data mining. In: *Proceedings of the First International Conference on Information Technology and Quantitative Management, ITQM 2013, 16–18 May 2013, Dushu Lake Hotel, Sushou, China.* 104–113.

**Kiayias A, Neumann J, Walluck D, McCusker O. 2009.** A combined fusion and data mining framework for the detection of botnets. In: *2009 Cybersecurity Applications & Technology Conference for Homeland Security.* IEEE, 273–284.

**Kisilevich S, Keim DA, Rokach L. 2013.** A gis-based decision support system for hotel room rate estimation and temporal price prediction: the hotel brokers' context. *Decision Support Systems* **54(2)**:1119–1133 DOI 10.1016/j.dss.2012.10.038.

**Kühn A, Joppen R, Reinhart F, Röltgen D, Von Enzberg S, Dumitrescu R. 2018.** Analytics canvas-a framework for the design and specification of data analytics projects. *Procedia CIRP* **70**:162–167 DOI 10.1016/j.procir.2018.02.031.

**Kumar A, Shankar R, Choudhary A, Thakur LS. 2016.** A big data mapreduce framework for fault diagnosis in cloud-based manufacturing. *International Journal of Production Research* **54(23)**:7060–7073 DOI 10.1080/00207543.2016.1153166.

**Kumar A, Shankar R, Thakur LS. 2018.** A big data driven sustainable manufacturing framework for condition-based maintenance prediction. *Journal of Computational Science* **27**:428–439 DOI 10.1016/j.jocs.2017.06.006.

**Kurgan LA, Musilek P. 2006.** A survey of knowledge discovery and data mining process models. *Knowledge Engineering Review* **21(1)**:1–24 DOI 10.1017/S0269888906000737.

**Lau RYK, Zhang W, Xu W. 2018.** Parallel aspect-oriented sentiment analysis for sales forecasting with big data. *Production and Operations Management* **27(10)**:1775–1794 DOI 10.1111/poms.12737.

**Lawler J, Joseph A. 2017.** Big data analytics methodology in the financial industry. *Information Systems Education Journal* **15(4)**:38.

**Lee W, Nimbalkar RA, Yee KK, Patil SB, Desai PH, Tran TT, Stolfo SJ. 2000.** A data mining and CIDF based approach for detecting novel and distributed intrusions. In: *Third International Workshop, Recent Advances in Intrusion Detection, RAID 2000, 2–4 October 2000, Toulouse, France.* 49–65.

**Lee W, Stolfo SJ, Chan PK, Eskin E, Fan W, Miller M, Hershkop S, Zhang J. 2001.** Real time data mining-based intrusion detection. In: *Proceedings DARPA Information Survivability Conference and Exposition II.* IEEE, 89–100.

**Lee W, Stolfo SJ, Mok KW. 1999.** Mining in a data-flow environment: experience in network intrusion detection. In: *Proceedings of the Fifth ACM SIGKDD International Conference on Knowledge Discovery and Data Mining, 15–18 August 1999, San Diego, CA, USA.* 114–124.

**Lemieux VL. 2016.** Innovating good regulatory practice using mixed-initiative social media analytics and visualization. In: *2016 Conference for E-Democracy and Open Government, CeDEM 2016, 18–20 May 2016, Krems, Austria.* 207–212.

**Lenz J, Wuest T, Westkämper E. 2018.** Holistic approach to machine tool data analytics. *Journal of Manufacturing Systems* **48**:180–191 DOI 10.1016/j.jmsy.2018.03.003.

**Létourneau S, Yang C, Drummond C, Scarlett E, Valdés J, Zaluski M. 2005.** A domain independent data mining methodology for prognostics. In: *Essential Technologies for Successful Prognostics: Proceedings of the 59th Meeting of the Society for Machinery Failure Prevention Technology, 18–21 April 2005, Virginia Beach, Virginia.*

**Liu F, Xu R, Fan W, Jiang Z. 2018.** Data analytics approach for train timetable performance measures using automatic train supervision data. *IET Intelligent Transport Systems* **12(7)**:568–577 DOI 10.1049/iet-its.2017.0287.

**Lu Q, Lyu Z-J, Xiang Q, Zhou Y, Bao J. 2017.** Research on data mining service and its application case in complex industrial process. In: *2017 13th IEEE Conference on Automation Science and Engineering (CASE).* IEEE, 1124–1129.

**Luna JM, Castro C, Romero C. 2017.** MDM tool: a data mining framework integrated into moodle. *Computer Applications in Engineering Education* **25(1)**:90–102 DOI 10.1002/cae.21782.

**Mahmood A, Shi K, Khatoon S, Xiao M. 2013.** Data mining techniques for wireless sensor networks: a survey. *International Journal of Distributed Sensor Networks* **9(7)**:406316 DOI 10.1155/2013/406316.

**Marbán O, Mariscal G, Ruiz EM, Segovia J. 2007.** An engineering approach to data mining projects. In: *8th International Conference, Intelligent Data Engineering and Automated Learning - IDEAL 2007, 16–19 December 2007, Birmingham, UK.* 578–588.

**Marban O, Mariscal G, Segovia J. 2009.** A data mining and knowledge discovery process model. In: Julio P, Adem K, eds. *Data Mining and Knowledge Discovery in Real Life Applications.* Paris: I-Tech, 438–453.

**Marbán O, Segovia J, Menasalvas E, Fernández-Baizán C. 2009.** Toward data mining engineering: a software engineering approach. *Information Systems* **34(1)**:87–107 DOI 10.1016/j.is.2008.04.003.

**Mariscal G, Marbán Ó, Fernández C. 2010.** A survey of data mining and knowledge discovery process models and methodologies. *Knowledge Engineering Review* **25(2)**:137–166 DOI 10.1017/S0269888910000032.

**Mobasher B. 2007.** *Data mining for web personalization: the adaptive web, methods and strategies of web personalization.* Berlin: Springer-Verlag, 90–135.

**Murnion P, Helfert M. 2011.** A framework for decision support for learning management systems. In: *10th European Conference on e-Learning ECEL-2011, Brighton, UK.*

**Nestorov S, Jukic N. 2003.** Ad-hoc association-rule mining within the data warehouse. In: *36th Hawaii International Conference on System Sciences (HICSS-36 2003), CD-ROM/Abstracts Proceedings, 6–9 January 2003, Big Island, HI, USA.* 232.

**Niesen T, Houy C, Fettke P, Loos P. 2016.** Towards an integrative big data analysis framework for data-driven risk management in industry 4.0. In: *49th Hawaii International Conference on System Sciences, HICSS 2016, 5–8 January 2016, Koloa, HI, USA.* 5065–5074.

**Nohuddin P, Zainol Z, Lee ASH, Nordin I, Yusoff Z. 2018.** A case study in knowledge acquisition for logistic cargo distribution data mining framework. *International Journal of Advanced and Applied Sciences* **5(1)**:8–14 DOI 10.21833/ijaas.2018.01.002.

**Ortega JP, Iturbide E, Olivares V, Hidalgo MA, Almanza N, Rebollar AM. 2015.** A data preparation methodology in data mining applied to mortality population databases. In: Rocha A, Correia AM, Costanzo S, Reis LP, eds. *New Contributions in Information Systems and Technologies.* Vol. 1. Cham: Springer, 1173–1182.

**Osman AMS, Elragal A, Bergvall-Kåreborn B. 2017.** Big data analytics and smart cities: a loose or tight couple? In: *10th International Conference on Connected Smart Cities 2017 (CSC 2017), 20–22 July 2017.* Lisbon: IADIS, 157–168.

**Park G, Chung L, Zhao L, Supakkul S. 2017.** A goal-oriented big data analytics framework for aligning with business. In: *Third IEEE International Conference on Big Data Computing Service and Applications, BigDataService 2017, 6–9 April 2017, Redwood City, CA, USA.* 31–40.

**Pournaras E, Nikolic J, Velásquez P, Trovati M, Bessis N, Helbing D. 2016.** Self-regulatory information sharing in participatory social sensing. *EPJ Data Science* **5(1)**:14 DOI 10.1140/epjds/s13688-016-0074-4.

**Pouyanfar S, Chen S. 2016.** Semantic concept detection using weighted discretization multiple correspondence analysis for disaster information management. In: *17th IEEE International Conference on Information Reuse and Integration, IRI 2016, 28–30 July 2016, Pittsburgh, PA, USA.* 556–564.

**Puthal D, Nepal S, Ranjan R, Chen J. 2016.** A secure big data stream analytics framework for disaster management on the cloud. In: *18th IEEE International Conference on High Performance Computing and Communications; 14th IEEE International Conference on Smart City; 2nd IEEE International Conference on Data Science and Systems, HPCC/SmartCity/DSS 2016, 12–14 December 2016, Sydney, Australia.* 1218–1225.

**Rahman FA, Desa MI, Wibowo A. 2011.** A review of kdd-data mining framework and its application in logistics and transportation. In: *The 7th International Conference on Networked Computing and Advanced Information Management.* IEEE, 175–180.

**Rendall R, Lu B, Castillo I, Chin S-T, Chiang LH, Reis MS. 2017.** A unifying and integrated framework for feature oriented analysis of batch processes. *Industrial & Engineering Chemistry Research* **56(30)**:8590–8605 DOI 10.1021/acs.iecr.6b04553.

**Reutterer T, Hornik K, March N, Gruber K. 2017.** A data mining framework for targeted category promotions. *Journal of Business Economics* **87(3)**:337–358 DOI 10.1007/s11573-016-0823-7.

**Segarra LL, Almalki H, Elabd J, Gonzalez J, Marczewski M, Alrasheed M, Rabelo L. 2016.** A framework for boosting revenue incorporating big data. *Journal of Innovation Management* **4(1)**:39–68 DOI 10.24840/2183-0606_004.001_0005.

**Shahbaz M, Masood SA, Shaheen M, Khan A. 2010.** Data mining methodology in perspective of manufacturing databases. *Journal of American Science* **6(11)**:999–1012.

**Shao Z, Liu J, Zhu X. 2008.** Image mining for generating ontology databases of geographical entities. In: *Proceedings of the 8th International Symposium on Spatial Accuracy Assessment in Natural Resources and Environmental Sciences.* Edgbaston: World Academic Union (Press).

**Sharma S, Osei-Bryson K. 2008.** Organization-ontology based framework for implementing the business understanding phase of data mining projects. In: *Proceedings of the 41st Hawaii International International Conference on Systems Science (HICSS-41 2008), 7–10 January 2008, Waikoloa, Big Island, HI, USA.* 77.

**Sharma S, Osei-Bryson K. 2009.** Framework for formal implementation of the business understanding phase of data mining projects. *Expert Systems with Applications* **36(2)**:4114–4124 DOI 10.1016/j.eswa.2008.03.021.

**Shin MS, Jeong KJ. 2005.** An alert data mining framework for network-based intrusion detection system. In: *6th International Workshop on Information Security Applications, WISA 2005, 22–24 August 2005, Jeju Island, Korea.* 38–53.

**Shrivastava S, Pal SN. 2017.** A big data analytics framework for enterprise service ecosystems in an e-governance scenario. In: *Proceedings of the 10th International Conference on Theory and Practice of Electronic Governance, ICEGOV 2017, 7–9 March 2017, New Delhi, India.* 5–11.

**Simoff SJ, Galloway J. 2008.** Visual discovery of network patterns of interaction between attributes. In: *Visual Data Mining - Theory, Techniques and Tools for Visual Analytics.* Springer, 172–195.

**Singh K, Guntuku SC, Thakur A, Hota C. 2014.** Big data analytics framework for peer-to-peer botnet detection using random forests. *Information Sciences* **278**:488–497 DOI 10.1016/j.ins.2014.03.066.

**Singh S, Prasad A, Srivastava K, Bhattacharya S. 2016.** A cellular logic array based data mining framework for object detection in video surveillance system. In: *2016 2nd International Conference on Next Generation Computing Technologies (NGCT).* IEEE, 719–724.

**Singh S, Vajirkar P, Lee Y. 2003.** Context-based data mining using ontologies. In: *22nd International Conference on Conceptual Modeling, ER 2003, 13–16 October 2003, Chicago, IL, USA.* 405–418.

**Smith KA, Willis RJ, Brooks M. 2000.** An analysis of customer retention and insurance claim patterns using data mining: a case study. *Journal of the Operational Research Society* **51(5)**:532–541 DOI 10.1057/palgrave.jors.2600941.

**Solarte J. 2002.** A proposed data mining methodology and its application to industrial engineering. Ph.D. thesis, University of Tennessee.

**Strohbach M, Ziekow H, Gazis V, Akiva N. 2015.** Towards a big data analytics framework for IoT and smart city applications. In: Xhafa F, Barolli L, Barolli A, Papajorgji P, eds. *Modeling and Processing for Next-generation Big-data Technologies.* Cham: Springer, 257–282.

**Sun J, Xu W, Ma J, Sun J. 2015.** Leverage RAF to find domain experts on research social network services: a big data analytics methodology with mapreduce framework. *International Journal of Production Economics* **165**:185–193 DOI 10.1016/j.ijpe.2014.12.038.

**Tavares R, Vieira R, Pedro L. 2017.** A preliminary proposal of a conceptual educational data mining framework for science education: Scientific competences development and self-regulated learning. In: *2017 International Symposium on Computers in Education (SIIE).* IEEE, 1–6.

**Torres P, Marques P, Marques H, Dionisio R, Alves T, Pereira L, Ribeiro J. 2017.** Data analytics for forecasting cell congestion on LTE networks. In: *Network Traffic Measurement and Analysis Conference, TMA 2017, 21–23 June 2017, Dublin, Ireland.* 1–6.

**Van Rooyen M, Simoff SJ. 2008.** A strategic analytics methodology. In: *Proceedings of the Third International Conference on Software and Data Technologies, Volume ISDM/ABF, ICSOFT, 2008, 5–8 July 2008, Porto, Portugal.* 20–28.

**Wang L. 2015.** Data mining in functional test content optimization. In: *The 20th Asia and South Pacific Design Automation Conference, ASP-DAC 2015, 19–22 January 2015, Chiba, Japan.* 308–315.

**Wang L. 2017.** Experience of data analytics in EDA and test—principles, promises, and challenges. *IEEE Transactions on Computer-Aided Design of Integrated Circuits and Systems* **36(6)**:885–898 DOI 10.1109/TCAD.2016.2621883.

**Wijayasekara D, Linda O, Manic M. 2011.** CAVE-SOM: immersive visual data mining using 3D self-organizing maps. In: *The 2011 International Joint Conference on Neural Networks, IJCNN 2011, 31 July–5 August 2011, San Jose, CA, USA*. 2471–2478.

**Xiang L. 2009a.** Context-aware data mining methodology for supply chain finance cooperative systems. In: *Fifth International Conference on Autonomic and Autonomous Systems, ICAS 2009, 20–25 April 2009, Valencia, Spain*. 301–306.

**Xiang L. 2009b.** Integrating context-aware and fuzzy rule to data mining model for supply chain finance cooperative systems. In: *The Fourth International Conference on Software Engineering Advances, ICSEA 2009, 20–25 September 2009, Porto, Portugal*. 471–476.

**Xu S, Qiu M. 2008.** A privacy preserved data mining framework for customer relationship management. *Journal of Relationship Marketing* **7(3)**:309–322 DOI 10.1080/15332660802417236.

**Yang L, Shi Z. 2010.** An efficient data mining framework on hadoop using java persistence API. In: *10th IEEE International Conference on Computer and Information Technology, CIT 2010, 29 June–1 July 2010, Bradford, West Yorkshire, UK*. 203–209.

**Yang Y, Zheng Z, Huang C, Li K, Dai H. 2016.** A novel hybrid data mining framework for credit evaluation. In: *12th International Conference on Collaborate Computing: Networking, Applications and Worksharing, CollaborateCom 2016, 10–11 November 2016, Beijing, China*. 16–26.

**Yi W, Teng F, Xu J. 2016.** Noval stream data mining framework under the background of big data. *Cybernetics and Information Technologies* **16(5)**:69–77 DOI 10.1515/cait-2016-0053.

**Ying Y, Yinghong W, Rong J, Liquan J. 2014.** Domain driven data mining for customer demand discovery. In: *2014 11th International Conference on Service Systems and Service Management (ICSSSM)*. IEEE, 1–6.

**Yu Z, Fung BC, Haghighat F. 2013.** Extracting knowledge from building-related data: a data mining framework. *Building Simulation* **6(2)**:207–222 DOI 10.1007/s12273-013-0117-8.

**Yuan B, Herbert J. 2014.** A cloud-based mobile data analytics framework: case study of activity recognition using smartphone. In: *2nd IEEE International Conference on Mobile Cloud Computing, Services, and Engineering, MobileCloud 2014, 8–11 April 2014, Oxford, United Kingdom*. 220–227.

**Yuan B, Herbert J, Emamian Y. 2014.** Smartphone-based activity recognition using hybrid classifier: utilizing cloud infrastructure for data analysis. In: *Proceedings of the 4th International Conference on Pervasive and Embedded Computing and Communication Systems, PECCS 2014, 7–9 January 2014, Lisbon, Portugal*. 14–23.

**Yun Z, Weihua L, Yang C. 2014.** Applying balanced scordcard strategic performance management to CRISP-DM. In: *2014 International Conference on Information Science, Electronics and Electrical Engineering, 26–28 April 2014, Sapporo, Japan*. 2009–2014.

**Zaghloul MM, Ali-Eldin A, Salem M. 2013.** Towards a self-service data analytics framework. *International Journal of Computer Applications* **80(9)**:41–48 DOI 10.5120/13893-1840.

**Zaki M, Sobh TS. 2005.** NCDS: data mining for discovering interesting network characteristics. *Information & Software Technology* **47(3)**:189–198 DOI 10.1016/j.infsof.2004.08.002.

**Zaluski M, Létourneau S, Bird J, Yang C. 2011.** Developing data mining-based prognostic models for CF-18 aircraft. *Journal of Engineering for Gas Turbines and Power* **133(10)**:101601 DOI 10.1115/1.4002812.

**Zhang Z. 2009.** An efficient neuro-fuzzy-genetic data mining framework based on computational intelligence. In: Yu G, Köppen M, Chen S, Niu X, eds. *9th International Conference on*

*Hybrid Intelligent Systems (HIS 2009), August 12–14, 2009, Shenyang, China.* Piscataway: IEEE, 178–183.

**Zhang W, Lau RYK, Li C. 2014.** Adaptive big data analytics for deceptive review detection in online social media. In: *Proceedings of the International Conference on Information Systems: Building a Better World through Information Systems, ICIS 2014, 14–17 December 2014, Auckland, New Zealand.*

**Zhao K, Liu B, Tirpak TM, Xiao W. 2005.** Opportunity map: a visualization framework for fast identification of actionable knowledge. In: *Proceedings of the 2005 ACM CIKM International Conference on Information and Knowledge Management, 31 October–5 November 2005, Bremen, Germany.* 60–67.

**Zhong RY, Xu C, Chen C, Huang GQ. 2017.** Big data analytics for physical internet-based intelligent manufacturing shop floors. *International Journal of Production Research* **55(9)**:2610–2621 DOI 10.1080/00207543.2015.1086037.