# Peer review of "Adaptations of data mining methodologies: a systematic literature review"

_PeerJ Computer Science, doi:10.7717/peerj-cs.267_

## Round 0.1 · original submission · Major Revisions

Please follow the recommendations given for the reviewer, and I think they can improve the paper substantially. Pay special attention to the replicability of the review performed.

Reviewer 1 ·

Basic reporting

The paper reviews the application of data mining methodologies/processes, either “as-is” or adapted for some specific use. The authors have followed the SLR (systematic literature review) method suggested by Kitchenham et al., which actually is the most widely accepted method for compiling, identifying, extracting and analysing studies on a given topic.

The review topic is not especially novel itself, as there is a large number of SLRs and surveys on the area, focused on different but related aspects. In contrast, the focus that the authors have tried to give to this paper is relevant and interesting for both the academia and industry. Unfortunately, the paper has some weak points that should be considered.

Introduction:
- Footnote 1. Unfortunately, I cannot agree with the authors when claiming that they use the terms “data mining” and “data analytics” as synonyms. It is broadly accepted that both terms are rather different, and, in the case of the paper, it might cause confusion and affect readability given that DM is itself a phase e.g. within the KDD process. Please rewrite where necessary to make this different explicit. This is, in fact, mentioned by the authors in line 122 (Background), where they need to disambiguate its use.

- Murnion and Helfert (2011) does not seem to be most appropriate reference for introducing the EDM area. I suggest the following cite instead:
Romero, C., Ventura, S. “Data Mining in Education”. Wiley Interdiscip. Rev. Data Min. Knowl. Discov. 3(1):12-27, 2013; DOI:10.1002/widm.1075

- Footnote 2. The reference (Kitchenham, 2004) is relevant enough to be inserted as a cite within the text, not as a footnote. A more recent and generalizable reference could be the following:
Kitchenham, B.A., Budgen, D., Brereton, P. “Evidence-based software engineering and systematic reviews”. CRC Press, 2016.

- Footnote 4. The use of this type of non-scientific, validated, peer-reviewed cites should be extremely limited. The required formality, precision and experimental replicability is not contrasted or guaranteed in this article, even when KDNuggets is a valuable blog for the community. I do not think that the authors could use a post, though interesting, as a way to motivate a research gap that justify their paper. This is a weak point for revision.

- Line 95. Please define from the beginning the term “grey literature”, and make explicit the scope limitation of these studies for this review.


Background:
- Footnote 5 (and others). Please reference web sites correctly including at least [if available: authors, date], title, URL and last access.

- Figure 2 is important for the review and interesting for the reader. Surprisingly, no references after 2008 are considered (more than 10 years ago), as the figure is directly extracted from (Mariscal et al., 2010). Given the topic of this paper, and the need to consider recent methodologies and processes, this figure should be properly completed for this study, for instance, to consider new approaches more focused on recent approaches, platforms and technologies like Big Data (such as the paper itself begins).

- Related to the latest point above, a major concern with this paper, and later reflected in the methodology, is that the list of considered frameworks seems to be outdated from the beginning (otherwise, it is not properly justified the reason why no other process is considered). In fact, later, the authors mostly focus the review on CRISP-DM. I agree this is the most popular process, but -considering what the SLR seems to promise- it does not have to be the only one considered, not even assume that it is the majority. All this information could effect the search. According to Table 1, ASUM is apparently the only model since 2007. If so, any reason? Besides, this makes necessary to justify the need to this SLR itself, given the lack of model developments in the field (with some exception).


Research design:
Even though I will go deeper into this section later in the "Study design", let me please comment on some general aspect on this point.

- Sometimes the paper becomes somewhat verbose, in the sense that it explains the same aspects more than once. A clear example is the research questions, which appear at least 4 times in the paper (!) At this point, it does not help readability. Please make a critical reading to avoid it. For example, I would remove RQs from the abstract, and explain the objectives in the introduction (it does not mean to explicitly write them there), as they are introduced and detailed in the section “Research questions”. This is just a suggestion. The same happens when e.g. talking about the adaptations to methodologies and models (e.g. text between 495-499 has no specific contribution).

- Line 339. Please clarify what a “version” is. To what extend has a paper to be altered, extended, updated or rewritten to be considered as a new paper, instead of a version?

- Line 347. Screening steps (Figure 4, end of page 11) should be explained before being referenced in Table 2 (page 10).

- Figure 4. It seems odd that the summary is evaluated in step 6, and not in step 4, as expected. Can a title be significant enough to exclude a paper? It seems reasonable to consider both the title and the abstract in step 4 (the abstract is provided by most databases), while step 6 focuses on other sections such as the introduction or conclusions. This is a common practice, in fact. This point affects the methodology, so could be included in section “Study design” of this review too.

- Figure 4. “599” is the number of candidate papers, and “207” the number of primary studies, right?

- Lines 379 to 393 (including Figure 5). This should not be part of the Research design. Instead, I would suggest adding a new section called “Quantitative analysis” or “Quantitative results” in order to fully explain more numerical information of interest about the findings, e.g. number of authors, number of institutions, rate industry/academia, type of uses, etc.

- “Grey literature” has to be uniquely defined. For example, in line 391, “articles in moderated and non-peer reviewed journals” are referred. How can we know that this literature meets the minimum quality criteria to be considered as part of the SLR? This type of (“grey”) literature requires attention, since it should be fully verifiable, follow the scientific method with a minimum of quality, etc. How can we ensure it all in the case of non-peer reviewed publications? For instance, have predatory journals been considered? Again, this point affects the SLR method, and (as currently justified) I cannot agree with including this type of papers. Please clarify this point.

- Line 398. Threats to the validity should be presented as a separate section. If so, the reference to this section in this line should be removed.

- Figure 6. Why were 1997-2007 and 2008-2018 chosen as periods? Is there any disruptive/significant event that could imply a more relevant cut?


Findings and discussion:
- A major issue of the paper is the lack of evidence. The authors mention some relevant facts that are not easily verifiable. For instance, in line 414, they mention that their “review led us to identify three distinct adaptation scenarios”. How? Why? Where are these scenarios from? Are there any other minor scenarios that could be discarded? Were they obtained from the literature or predetermined beforehand? A previous detailed quantitative analysis endorsing the following decisions could help the reader to understand why the authors made one decision or another. Without evidence, any point would become speculative.

- Again, in line 522, it is said that “we noted that ‘extension’ to…”. Please provide some evidence on findings.

- Figure 7. As a reader, I would appreciate to know more about how the adaptation scenarios have been conducted by methodological model (CRISP-DM, KDD…).

- With respect to RQ2, I miss a further analysis on the use, types of frameworks and industrial opportunities. The latter is, in fact, something I would suggest reinforcing and discussing in further detail. How industry has adopted/adapted different models? Benefits? Lessons learnt?

- Along the paper (more particularly in this section), it seems that CRISP-DM is used as baseline. What about other models? This is never justified (and it is marginally related to the comment above about dates and models). This point is especially notorious when responding to RQ3 (e.g. lines 494, 708, 770).

- Line 489-520. I am aware that it is a SLR, not a survey. But, in any case, it remains unclear how these modifications are really applied. To what extend could they be extended or extrapolated to other uses?

- In my particular opinion, the paper improves its readability and analysis from line 538 (subsection “Extension").

- I would suggest considering a greater difference between industrial and academic studies, industrial cases and simulations/theoretical cases. This information would be valuable to infer how these proposals have actually affected real projects and developments.

- I cannot see the contribution of subsection “Summary”, mostly referred to lines 754 to 768.


General issues and minor comments:
- In line 45, what do you mean by “in the same year”? What year, 2016?

- Please check that items in lists are consistently ended with a period (e.g. lines 129, 131, 135, 138, 143, 146, 148, 195, 198, 200, 206, etc.)

- Figures are low quality. Please re-render figures with acceptable quality. Some are difficult to read. Some also replicate the same misspelling (Figs. 9-11).

Experimental design

A strong weakness of the paper is related to the lack of replicability. This is the base of any SLR, and should be carefully considered. Unfortunately, I would not recommend this paper for acceptance if this point is not properly addressed.

- Methodology is not complete:
1. Snowballing is not conducted, or explained.
2. “Data extraction” should be a subsection itself. The authors should explain the data extraction form (conformant to the form provided as additional material), the exhaustive method to extract these data, responsibilities among authors, how conflicts were faced, etc.
3. The authors mention the SLR protocol (e.g. line 785), but the document is not available as additional material, or for download. Please provide the protocol document, which is required for replicability.

Other general and/or minor comments related to the methodology:
- Line 283. Why are terms like “process” or “process model” omitted from the search? They seem to be related with the improvement and modification of frameworks.

- There is a limited use of databases. Even when the authors try to explain why they have not used other databases like ACM, ScienceDirect or IEEExplore, the reason is not really convincing without previous pilot searches that show such a correspondence. Further evidence is required or just adding these databases.

- Why did the authors used Google Scholar, but they did not try Arxiv for grey literature?

Validity of the findings

- Structure. I would suggest adding a new section for discussion on open issues, found gaps and trends in the field, as extracted from the primary studies.

- Structure. “Threats to the validity” should be a section itself, not a subsection. (Sorry if it is like this now, but it is confusing the format without numbering sections).

Additional comments

- I suggest creating two distinct sections for the bibliography: “REFERENCES” and “PRIMARY STUDIES”. This is a common practice in SLR, and much more intuitive and readable.

- Please check references. For example, Forbes(2017) is incomplete: author’s name, title, date, etc.

---

## Round 0.2 · Minor Revisions

Please follow the recommendations given by the reviewer.

Reviewer 2 ·

Basic reporting

This submission is a revised version, in which the authors seem to have dealt with most of the concerns of the reviewers. I am seeing this paper for the first time.

This paper has provided a review of different data mining methodologies. Overall, the paper is well written and has covered well the current state-of-the-art on data mining methodologies. Although there are already many review papers on the topic, I believe that the way that the authors are discussing data mining methodologies is novel and very interesting.

Experimental design

The design of the study is correct from my point of view.

Validity of the findings

The main conclusions and findings of this work are certainly valid, and well supported by the literature.

Additional comments

- I believe that it is worth including a note on machine learning techniques and their differences with data mining.
- Page 3 – line 122. I disagree with the term big data. Data analytics does not need to analyse big datasets, it could be small datasets as well. Please consider rewording this sentence.
- Page 16 – line 510. I am not sure I agree with the KDD processes being changed because of Big Data. The workflow remains the same, but there are differences (other challenges) to be addressed in terms of preprocessing, gathering information, storing information, and mining the data. The reader would benefit from some related references in this regard.
- Be careful with the use of double quotes, they seem to be incorrect.
- Modelling vs modelling (please stick to American English).

---

## Round 0.3 · accepted · Accept

Congratulations on the acceptance.